# CorrectionPlanner: Self-Correction Planner with Reinforcement Learning in Autonomous Driving

**Yihong Guo** [1]  **Dongqiangzi Ye** [2]  **Sijia Chen** [2]  **Anqi Liu** [1]  **Xianming Liu** [2]

## Abstract

Autonomous driving requires safe planning, but most learning-based planners lack explicit self-correction ability: once an unsafe action is proposed, there is no mechanism to correct it. Thus, we propose CorrectionPlanner, an autoregressive planner with self-correction that contains a propose, evaluate, and correct loop in the motion-token generation process. At each planning step, the policy proposes an action, namely a motion token, and a learned collision critic predicts whether it will induce a collision within a short horizon. If the critic predicts a collision, we retain the sequence of historical unsafe motion tokens as a self-correction trace, generate the next motion token conditioned on it, and repeat this process until the safe motion token is proposed or the safety criterion is met. This self-correction trace, consisting of all the unsafe motion tokens, represents the planner's correction process in motion-token space. We train the planner with imitation learning followed by model-based reinforcement learning using rollouts from a pretrained world model that realistically models agents' reactive behaviors. Closed-loop evaluations show that CorrectionPlanner reduces the collision rate by over $20\%$ on Waymax and obtains state-of-the-art planning scores on nuPlan.

## 1. Introduction

Autonomous driving (Inamdar et al., 2024) is a long-horizon decision-making problem that requires both efficiency and safety. Most learning-based planners learn rich scene context, including agents, maps, and navigation information,

Work done during an internship at XPENG Motors. [1]Department of Computer Science, Johns Hopkins University, Baltimore, USA [2]XPENG Motors. Correspondence to: Yihong Guo <yguo80@jhu.edu>.

*Proceedings of the 43$^{rd}$ International Conference on Machine Learning*, Seoul, South Korea. PMLR 306, 2026. Copyright 2026 by the author(s).

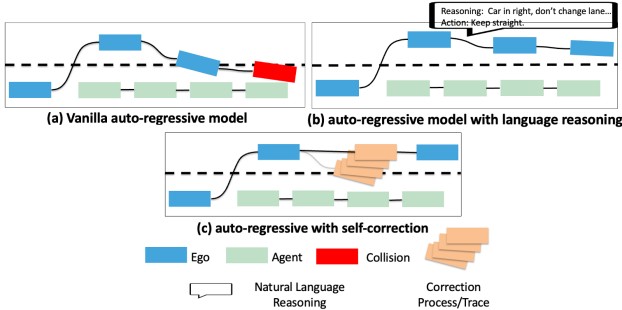

*Figure 1.* (a) Vanilla autoregressive models, no correction mechanism. It does not have the revisit and correction step before deployment, leading to more unsafe behavior. (b) Autoregressive models that generate tokens sequentially, with reasoning through language. Though effective, such reasoning can be inefficient and does not have explicit self-correction before deployment. (c) Autoregressive models that generate subsequent tokens sequentially with self-correction through motion tokens. Our method explicitly corrects unsafe behavior before deployment with a correction trace, thereby reducing unsafe behavior.

and predict the next ego action or the trajectory (Wu et al., 2024; Zheng et al., 2025) as shown in Figure 1 (a). Also, recent work in LLMs (Havrilla et al., 2024) shows that explicit reasoning and self-reflection (Ferrag et al., 2025) can improve the model's reliability and avoid unsafe and hazardous output. Such methods have been widely adopted in autonomous driving, where reasoning is typically performed in language space, as illustrated in Figure 1(b). While effective, these models often lack an internal mechanism to evaluate and correct unsafe actions before execution.

Inspired by these, we propose an autonomous driving planner, CorrectionPlanner, with an explicit correction mechanism before execution that autoregressively corrects unsafe actions until the safety criterion is met. Instead of reasoning in natural language, our method performs self-correction at the motion token level, discretized from trajectories. Specifically, at each planning step, the policy (planner) first autoregressively generates an ego motion token given the agents'/ego's history, map/navigation information, and predicted agent motions, and then a learned collision critic evaluates whether it will cause a collision within a short horizon (e.g., 2.5s). If it is unsafe, the token will not be executed. The policy retains the sequence of all unsafe

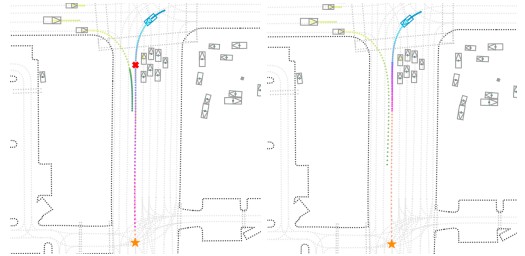

*Figure 2.* Blue box is ego; red cross is collision; orange trajectory is the expert trajectory. Left: autoregressive without self-correction. Right: autoregressive with self-correction. Collision is avoided with self-correction before the collision by slowing down to yield to the right-turn vehicle and then re-accelerating.

motion tokens and repeatedly generates new tokens based on this sequence. Such a sequence of historically proposed unsafe actions, which we call the correction trace, includes all tokens generated during the correction process and serves as the planner's internal *correction trace* in motion-token space, analogous to the *reasoning trace* in language models. We illustrate this mechanism in Figure 1(c). As shown in Figure 2, such a self-correction process can avoid the collision without relying on post-hoc filtering, rule-based candidate selection, or language reasoning.

To enable the planner to learn not only how to imitate expert behavior but also how to recover and self-correct from unsafe behaviors, we use a two-stage training pipeline including imitation learning and reinforcement learning. Imitation learning via next-token prediction trains only on expert trajectories, and no self-correction is applied. As a result, it does not teach the planner how to revise unsafe actions. We therefore adopt a two-stage training scheme: (1) imitation learning with next-token prediction on expert trajectories, and (2) model-based reinforcement learning using a pre-trained frozen world model (Wu et al., 2024) to simulate reactive multi-agent rollouts. During RL, the policy generates both motion tokens and correction trace tokens through the self-correction loop, and we optimize the ego policy with a safety-aware reward using a REINFORCE objective with KL regularization.

In summary, our contributions are:

1. We propose *CorrectionPlanner*, an autoregressive motion-token generation planner with a propose, evaluate, and correct loop that uses the *correction trace* to iteratively refine unsafe actions before execution.
2. We introduce a two-stage training scheme: imitation learning and model-based RL with a reactive multi-agent world model, enabling realistic rollouts that include both executed ego tokens, correction traces, and agents' reactive behavior.
3. Our approach reduces collision rates through self-correction, outperforming learning-based planners by over 20% in collision rate on the WOMD and achiev-

ing state-of-the-art planning scores on nuPlan datasets. And qualitative results show that our method exhibits diverse self-correction behaviors.

## 2. Related Work

**Motion prediction and planning.** Learning-based trajectory planning methods have been widely studied, including continuous motion distribution prediction (Bansal et al., 2018; Renz et al., 2022; Nayakanti et al., 2022; Huang et al., 2023; Cheng et al., 2023; 2024b; Cusumano-Towner et al., 2025; Xiao et al., 2025), diffusion-based models (Scheel et al., 2022; Hu et al., 2024; Wang et al., 2024; Zheng et al., 2025), and autoregressive models (Zhou et al., 2024; Wu et al., 2024). Though many of these works develop useful techniques for learning representations and achieving strong performance in simulation, they lack the ability to reflect on the actions before execution, which can lead to unsafe trajectories. In contrast, our method trained with RL has the ability to self-correct and refine the proposed motion token in an autoregressive manner, avoiding unsafe trajectories.

**Safe motion planning.** Safe motion planning with formal guarantee in robot manipulation tasks (Nakamura et al., 2025; Ak et al., 2023; Ke et al., 2024) and autonomous driving (Leung et al., 2023; Ma et al., 2026) is another line of work that is related to our method. However, they are mostly designed for more abstract or simplified environments to achieve a safety guarantee, while we study a more realistic, complex, and data-driven problem that evaluates general planning ability beyond safety. For example, reachability (Shao et al., 2021; Nakamura et al., 2025) and safety-filtering methods (Pek & Althoff, 2020) assume known or structured system dynamics, explicit safety geometry, or reachable/invariant set computation under strong assumptions on the dynamics. Besides, these methods are primarily designed around safety-oriented formulations for planning and are particularly useful when the objective is to guarantee constraint satisfaction under well-specified assumptions. However, the assumption or formulation makes it challenging to directly scale to realistic, large-scale, multi-agent driving settings, where the planner must remain both safe and behaviorally effective, such as efficiently following the navigation and progressing. Our work takes a complementary approach: rather than framing planning as a safety-centric problem or relying on a separate safety filter/verification module, we incorporate a self-correction mechanism into the autoregressive planning framework to reduce collisions while preserving the planner's general driving capability.

**Reinforcement learning for autonomous driving.** Reinforcement learning has been widely applied in autonomous driving, either followed by imitation learning (Lu et al., 2023; Huang et al., 2025) or pure RL (Scheel et al., 2022; Li et al., 2024; Zhang et al., 2025a). In these works, the

RL mostly serves as a post-training or fine-tuning step to improve the model's performance beyond the expert data. In contrast, our method uses RL to endow the model with self-correction ability, using rollout trajectories with correction tokens and rule-based rewards through two-stage training. Further, traditional RL methods like REINFORCE (Zhang et al., 2021), SAC (Haarnoja et al., 2018), and GRPO (Shao et al., 2024) are widely used in the literature, and we select REINFORCE with KL divergence to fine-tune the imitation learning model.

**Reasoning and self-reflection**. Reasoning (Zhang et al., 2024; Ferrag et al., 2025), self-reflection (Ji et al., 2023; Kamoi et al., 2024), latent reasoning (Zhu et al., 2025), and visual planning (Xu et al., 2025) are effective in LLMs for solving math/coding problems, and safety alignment (Zhang et al., 2025b), etc. Such methods mostly perform reasoning and self-reflection in the language space, i.e., using human language to represent the reasoning trace. However, using human language as a medium might not be optimal, as it may not accurately reflect the physical world or lead to redundant reasoning. Inspired by this, our proposed method performs self-correction using motion tokens and generates actions via correction traces to reduce collision rates.

# 3. Preliminaries

## 3.1. Self-Correction Autonomous Driving as Model-Based Reinforcement Learning

We model autonomous driving planning as a finite-horizon Markov decision process (MDP) with: $\mathcal{M} = (\mathcal{S}, \mathcal{A}, P, r, \gamma, H)$. $\mathcal{S}$ is the state space: $s_t = (s_{<t}^{\text{ego}}, s_{\leq t}^{\text{agents}}, m)$, where $s_t^{\text{ego}}$ includes the ego's information like position, speed and $m$ is the map and navigation information. $\mathcal{A}$ is the ego action space and action is the next motion token or position $s_t^{\text{ego}}$, $P$ is the transition kernel: $P(s_{t+1}|s_t, s_t^{\text{ego}})$ that modeling agent's behaviors given the planning command of the ego, $r$ is the reward function, $\gamma \in (0, 1]$ is a discount factor, and $H$ is the horizon. We parameterize the policy as $\pi_\theta(s_t^{\text{ego}} \mid s_{<t}^{\text{ego}}, s_{<t}^{\text{agents}}, \hat{s}_t^{\text{agents}}, m)$, where we output the next motion tokens $s_t^{\text{ego}} \in \mathcal{A}$ (e.g., a motion token or low-level control command), using the state information $s_t$ as well as the *predicted next state of the agents* using the learned world model $\hat{P}$. More generally, in the self-correction scenario, we further express the policy as: $\pi_\theta(s_t^{\text{ego}} \mid s_{<t}^{\text{ego}}, s_{<t}^{\text{agents}}, \hat{s}_t^{\text{agents}}, m, s_{t;<c}^{\text{ego}})$, where $s_{t;<c}^{\text{ego}}$ is the proposed unsafe motion token sequence (correction trace). Then, the environment evolves according to $P$, and the ego receives a scalar reward $r(s_t, s_t^{\text{ego}})$.

The goal is to learn a policy that maximizes expected reward: $J(\pi_\theta) = \mathbb{E}\left[\sum_{t=1}^{H} \gamma^t r(s_t, s_t^{\text{ego}})\right]$, where the expectation is over the initial state distribution, the environment dynamics

$P$, and the policy $\pi_\theta$.

**Model-Based Reinforcement Learning with World Model**. A key challenge for RL training in autonomous driving is how to realistically simulate agents' behaviors given the ego vehicle's actions. Naive solutions include 1) replay log trajectories for the agents and 2) predict the future trajectories for all the timesteps first, and then plan the ego, like diffusion planner (Zheng et al., 2025). However, both are nonreactive simulations and ignore the ego's interventions in the agents, which can be unrealistic, especially when the ego performs self-correction, as the correction might further change the agents' actual behavior. We address this issue by using a *model-based reinforcement learning method*, where we learn the transition kernel $\hat{P}(s_{t+1}|s_t, s_t^{\text{ego}})$, also called the world model, which can reactively model the agent's behavior. This enables us to collect more realistic trajectories for the RL training.

## 3.2. Learned World Model for Multi-Agent Dynamics

We build our world model on top of SMART (Wu et al., 2024), a vectorized autoregressive multi-agent world model that generates reactive agents' behaviors.

**Tokenization.** We discretize continuous trajectories into motion tokens by segmenting them into 0.5-second intervals and clustering motion segments using a K-disk algorithm based on their average corner distance. This yields a discrete 1024-token vocabulary, where each token encodes a one-step change in position and heading. The agent's trajectory is represented as a token sequence over the horizon, allowing us to model the planning as a next-token generation problem.

**Architecture.** The world model is a transformer decoder with spatio-temporal attention as shown in Figure 3 (a). Motion tokens are first fused with vehicle information, including speed, size, and other factors, and then processed by six attention blocks. Each block consists of three components: (1) Temporal self-attention, modeling motion continuity and capturing each agent's long-range dependencies across the time horizon; (2) Agent-map cross-attention, which incorporates spatial constraints by attending to road elements, ensuring adherence to the road network, and reducing off-road behavior; (3) Agent-agent cross-attention, which models interactions with surrounding agents to simultaneously predict the future behavior of agents. To maintain rotation and translation invariance, relative spatio-temporal position embeddings and the distances between agents, the ego, and map tokens are applied to all attention modules. Both ego and non-ego agents share the same encoder structure, and the token vocabulary is shared between them.

**Loss.** The model is trained with a next-token prediction objective with cross-entropy loss for all agents:

$$\mathcal{L}_{\text{world}}(\phi) = -\sum_{t=1}^{H} \sum_{n=0}^{N} \log p_\phi(s_t^n \mid s_{<t}^{0:N}, m),$$

where $H$ is the horizon, $N$ is the number of agents, $s_t^n$ is the n-th agent's state at $t$, $s_{<t}^{0:N}$ is all the agents' token before $t$ and $m$ is the map information. Once pretrained, the world model is frozen during later RL training and used as a "simulator" within our RL loop to reactively sample trajectories for policy optimization and self-correction. We summarize the training of the world model in Algorithm 1.

# 4. Method

In this section, we present our method for detecting unsafe motions and refining them through an iterative correction process with the correction trace. During planning, each proposed motion token is evaluated by a collision critic that predicts whether it will cause a collision within a short horizon. If a proposed motion token is flagged as unsafe, the policy will not execute it, but it does not resample from the original token distribution. Instead, it appends the rejected token to a correction trace, the sequence of all previously rejected unsafe motion tokens, and generates a revised motion token conditioned on the scene history and this trace, repeating until the critic predicts no collision or the maximum correction length is reached. We first describe the policy network for generating ego motions, then show how we conduct self-correction, and finally introduce our two-stage training scheme: imitation learning using expert data followed by RL to strengthen corrective behavior. The architecture is shown in Figure 3. We summarize imitation learning, RL training, and inference in Algorithms 2 to 4.

## 4.1. Policy Network

We design the policy network using a similar architecture to the world model, but only to model the ego's behavior, and include navigation information for planning. As shown in Figure 3 (a), the policy consists of a stack of cross-attention modules: (i) a route/navigation cross-attention layer, (ii) a temporal self-attention layer over the ego trajectory, (iii) an ego-map cross-attention layer that incorporates map context, and (iv) an ego-agent interaction layer that models the ego's interactions with other agents. Then the policy outputs a motion token distribution via an MLP prediction head and samples a motion from it. The policy is trained with the next-token prediction objective on expert trajectories:

$$\mathcal{L}_{\text{imitation}}(\theta) = -\sum_{t=1}^{T} \log \pi_\theta(s_t^{\text{ego}} \mid s_{<t}^{\text{ego}}, s_{<t}^{\text{agents}}, \hat{s}_t^{\text{agents}}, m),$$

where the token of the ego is predicted, conditioned on the ego/agent's historical behavior, maps, as well as the agent's predicted one-step future behavior.

**Correction tokens.** No separate token type is used for the self-correction, and *correction tokens are motion tokens* generated during the self-correction process. Unsafe intermediate proposed motions are *not executed*; they are retained in the input sequence as a *correction trace*. The

policy then generates the next proposal distribution using the correction trace, which is encoded by a self-attention layer, and the final accepted token is executed.

**Self-correction in imitation learning.** Although RL further strengthens self-correction, we train the imitation stage to *explicitly expose* the policy to collision scenarios, enabling it to learn to revise unsafe predictions during IL. At planning timestep $t$, if the policy proposal $\hat{s}_t^{\text{ego}}$ collides at $t$ (shown as the orange token in Figure 3 (a)), we generate corrected tokens through self-correction until no collision happens. Specifically, we retain all previously proposed (unsafe) tokens as a sequence, encode them with self-attention to obtain a fused representation, and generate the next corrected motion token $\hat{s}_{t,c+1}^{\text{ego}}$, conditioning the policy on all the previously proposed unsafe actions. The objective of the correction in IL is to minimize the cross-entropy loss between each proposed action in the correction trace and the expert trajectories:

$$\mathcal{L}_{\text{corr.}}(\theta) = -\sum_{t=1}^{T} \sum_{c=1}^{C} \log \pi_\theta(s_t^{\text{ego}} \mid s_{<t}^{\text{ego,agents}}, \hat{s}_t^{\text{agents}}, m, \hat{s}_{t,<c}^{\text{ego}}),$$

where $C$ is the maximum correction length, $\hat{s}_{t,<c}^{\text{ego}}$ is the ego's self-correction trace sampled before the correction step $c$. So, we force the policy to generate expert trajectories conditioned on the correction trace, thereby endowing it with preliminary self-correction capability. The **overall loss** of the IL phase is:

$$\mathcal{L}_{\text{pretrain}} = \mathcal{L}_{\text{imitation}}(\theta) + \mathcal{L}_{\text{corr.}}(\theta) + \mathcal{L}_{\text{world}}(\phi).$$

Here, we train the world model and the policy together. Note that in the IL stage, we do not enable the policy to correct the *future collision*, but only to correct the current collision. This is because the training uses expert data for next-token prediction; it is neither realistic nor ideal to correct the *expert trajectories*, and we cannot obtain the *future collision* or observe counterfactual futures under the expert trajectories without rollout in the imitation learning stage. That's why we require the world model to generate realistic trajectories and RL to empower the policy with self-correction ability.

## 4.2. Self-Correction with Reinforcement Learning

In this section, we describe the self-correction and reinforcement learning framework shown in Figure 3 (b). In the IL phase, we only perform next-token prediction and next-token correction. However, in real-world driving, the correction is not feasible when a collision will occur in the next planning step: for example, it is too late to brake if the next control action results in a collision, as they might already be close enough. The correction procedure needs to be done earlier. Thus, we propose using reinforcement learning to further train the policy and learn a collision critic that can correct unsafe motions in advance via self-correction.

**Rule-based Reward.** We define the trajectories' reward as $r = \text{Progression Rate} \cdot I(\text{Collision} = 0) - I(\text{Collision} = 1)$.

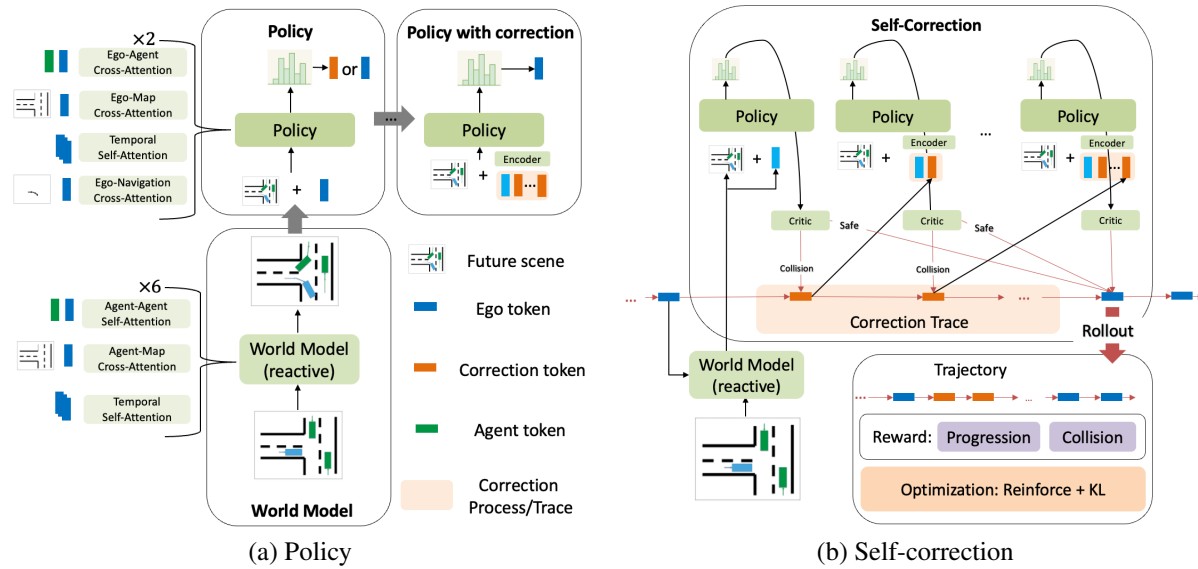

(a) Policy  (b) Self-correction

*Figure 3.* **(a) Policy architecture.** A frozen reactive world model predicts future motions for the ego and other agents. Conditioned on scene history and map/navigation context, the planner is an autoregressive policy that generates the next ego motion token. When self-correction is triggered, previously rejected proposals are retained as a *correction trace* and integrated via a self-attention encoder to condition subsequent proposals, providing an explicit correction signal during both imitation learning and RL. **(b) Self-correction.** At each planning timestep, the policy proposes an ego motion token, and a learned collision critic evaluates whether it will cause a collision within a short horizon. If unsafe, the policy iteratively generates revised motion tokens conditioned on the correction trace until a safe token is found (or a maximum number of correction steps is reached). The resulting rollout contains both executed ego tokens and intermediate correction tokens, and the policy is further optimized with REINFORCE using a rule-based reward and KL regularization.

Progression Rate is the progress relative to the expert trajectory, with a maximum of 1.0. We only use the collision and progression because 1) too many reward signals complicate the RL training, and 2) progression is one of the metrics that is most in conflict with collision, i.e., the planner might sacrifice the progression by moving very slowly to avoid collision, which is not ideal in the planning.

**Rollouts with World Model.** To enable reinforcement learning to improve self-correction beyond expert demonstrations, we generate rollouts using the pre-trained frozen world model. At each planning step, the ego policy first proposes a motion token, and the collision critic evaluates its collision risk within the horizon $k$. If the proposal is predicted to be unsafe, we reject it, append it to a *correction trace*, and autoregressively generate a revised motion token conditioned on the scene history and the entire correction trace, repeating until it becomes safe or the maximum correction length is reached. The final token is then executed in the world model, which generates the agents' behaviors. The model rollouts include executed ego tokens and the intermediate correction tokens in the correction trace.

**Policy Optimization.** We maximize the trajectory reward through REINFORCE and a KL divergence penalty:

$$\mathcal{L}_{\text{RL}}(\theta) = \sum_{t=1}^{T} r(\tau) \log \pi_\theta\left(s_{t,C_t}^{\text{ego}} \mid s_{<t}^{\text{ego}}, s_{\leq t}^{\text{agent}}, s_{t,<C_t}^{\text{ego}}, m\right)$$
$$- \lambda \text{KL}(\pi_{\text{imitation}} || \pi_\theta),$$

where $r(\tau)$ is the trajectory reward, $C_t$ denotes the cor-rection length (the length of the correction trace) at $t$, and $s_{t,<C_t}^{\text{ego}}$ is the correction trace consisting of all previously rejected ego motion tokens. $\lambda$ controls the scale of the KL regularization, which penalizes deviation from the IL policy. We apply the policy-gradient update only to the *executed* token $s_{t,C_t}^{\text{ego}}$, since the trajectory-level reward is attributed to the executed action but not the intermediate token in the correction trace. This objective learns to optimize the policy to generate a safe token given the entire correction trace.

### 4.3. Training the Collision Critic

To effectively detect future collisions for both RL training and inference, we train a critic (a binary classifier) to predict whether a collision will occur using rollout trajectory data from the policy and world model. The critic has two attention blocks and an MLP prediction head to predict the collision; each block contains: (i) a temporal self-attention layer over the ego and agents' trajectories, (ii) an ego-agent interaction layer that models the ego's interactions with other agents. The training objective is to minimize the cross-entropy loss:

$$\mathcal{L}_{\text{critic}}(\theta) = -\sum_{t=1}^{T} I(\text{collision}_{t:t+k} = 1) \log p_\theta\left(\text{collision} \mid s_{\leq t}\right)$$
$$+ I(\text{collision}_{t:t+k} = 0) \log p_\theta\left(\text{safe} \mid s_{\leq t}\right),$$

where $I(\text{collision}_{t:t+k} = 1)$ and $I(\text{collision}_{t:t+k} = 0)$ are collision or safe in $k$ future planning steps, respectively.

**Comparison with Rejection Sampling.** Although our self-correction iteratively generates motion tokens, it is *not equiv-*

*alent to rejection sampling under the safety constraints*. In rejection sampling, unsafe tokens are rejected, and the next proposal is *drawn from the same token distribution*. In contrast, our planner generates a new token distribution using information from the entire correction trace, enabling the policy to explicitly leverage it to update the distribution. Thus, the correction process changes the conditional distribution of future tokens based on past failures, analogous to multi-step "reasoning" in LLM but performed in motion-token space. We also compare against a rejection sampling baseline in Section 5.3 to demonstrate the difference.

## 5. Experiments

In this section, we evaluate our method on the WOMD (Ettinger et al., 2021) with the Waymax simulator (Gulino et al., 2023) powered by LatentDrive (Xiao et al., 2025) and the nuPlan dataset(Caesar et al., 2021). We also conduct various ablation studies to further validate our method. Our code is available at https://github.com/guoyihonggyh/CorrectionPlanner-Self-Correction-Planner-with-Reinforcement-Learning-in-Autonomous-Driving.

### 5.1. Experimental Setup

We evaluate our method in reactive and non-reactive modes: reactive agents follow an IDM policy (Treiber et al., 2000), while non-reactive agents use log-replay trajectories. For WOMD, we use the Waymax simulator (Xiao et al., 2025) powered by the LatentDrive (Xiao et al., 2025) to evaluate the closed-loop simulation. For nuPlan, evaluation is performed on the standard Val14, Test14-random, and Test14-hard splits (Dauner et al., 2023). More details can be found in Section B.1.

**Metrics.** For WOMD, we follow the simulator in Latent-Drive (Xiao et al., 2025) and report Off-Road, Collision, and Progression. For nuPlan, we report the planning score and Collision and Progression to validate that self-correction can reduce the collision rate.

### 5.2. Main Results

**Waymax Simulation.** In Table 1, we see that under both modes, our method achieves the **lowest collision rate**, reducing collisions by over 20% compared with the best baseline, demonstrating that self-correction is effective in reducing collisions. Meanwhile, our off-road performance remains consistently competitive, matching SMART baselines within a small margin, demonstrating that the spatio-attention mechanism is effective at modeling map-ego relationships and drivable areas. While LatentDrive has the highest progression rate, our method maintains a similar progression rate *while significantly improving safety*, sug-

gesting a favorable safety-efficiency trade-off.

**NuPlan Simulation.** In Table 2, we compare with the learning-based planners on nuPlan. Overall, our method shows consistently better performance across all settings except in the Test-Hard NR setting, where SMART performs the best. Further, our method outperforms the baselines and improves them more in the reactive mode, as observed in the Waymax experiments in Table 1, suggesting that reactive modeling of agents is more favorable in real-world interactive traffic. Further, we compare against the second-best baseline, SMART, on key metrics, including progress and collisions, shown in Table 3. Our method achieves a lower collision rate than SMART while maintaining similar performance on other metrics, demonstrating its effectiveness and a slight trade-off in those metrics, even though we only optimize the progression and safety reward in RL training.

**Why might progression decrease?** In many scenarios (e.g., left turns or crossing intersections), the self-correction planner often slows down or yields to other agents, and then re-accelerates afterward. This behavior effectively reduces collisions but can slightly harm the progression metric due to a temporary loss of speed. We believe this behavior is normal even for an expert driver and view it as an acceptable trade-off in autonomous driving, where safety should take precedence over marginal efficiency gains. Also, our model exhibits more complex and diverse self-correction behavior beyond slowing down/braking and yielding, as described in the qualitative results and the visualization in Section C.

**Qualitative Results.** Figure 4 visualizes three examples of our self-correction. The red cross means the collision. For (a) and (b), we see that the ego makes a slightly wider left turn in advance through self-correction to avoid the not-at-fault collision. Also, in (c) and (d), the ego is changing lanes, which might lead to a collision. So in (d), the self-correction corrects the aggressive lane-changing action to avoid a collision. In (e), the slow left turn leads to the collision; through self-correction, the ego avoids the braking and maintains a slightly higher speed to avoid a collision.

### 5.3. Ablation studies

We now present ablation studies and more baseline comparisons. In Section 5.3.1, we first show that the correction trace is beneficial for correcting the unsafe behavior compared to the variant without the correction trace. In Section 5.3.2, we also compare our method with different baselines, including the IL, RL without correction, etc., and we conduct an experiment on rejection sampling to further demonstrate that the self-correction cannot be achieved without our self-correction mechanism and the correction trace. In Section 5.3.3, we show the critic's precision and recall across different classification thresholds, along with the corresponding planning results, to study its correction

*Table 1.* Comparison with baselines on the Waymax simulator. Our method reduces the collision rate by over 20%. ∗ means we re-implement the method. We use **bold** and underline to denote the best and second best.

| Method | Reactive | | | Non-Reactive | | |
|---|---|---|---|---|---|---|
| | Collision ↓ | Offroad ↓ | Progression ↑ | Collision ↓ | Offroad ↓ | Progression ↑ |
| EasyChauffeur-PPO∗ (Xiao et al., 2024) | 4.71 | 3.81 | 97.93 | 4.53 | 2.14 | 96.42 |
| PlanT∗(Renz et al., 2022) | 2.94 | 1.65 | 95.85 | 3.02 | 1.43 | 95.48 |
| LatentDrive∗ (Xiao et al., 2025) | 3.27 | 2.33 | 98.70 | 3.11 | 1.94 | 99.21 |
| SMART∗ (Wu et al., 2024) | 2.36 | **0.87** | 91.33 | 4.30 | **0.86** | 90.87 |
| Ours | **1.68** | 0.94 | 94.23 | **2.43** | 0.92 | 95.75 |

*Table 2.* Comparison with baselines on nuPlan under non-reactive (NR) and reactive (R) settings. Our method outperforms baselines. ∗ means we re-implement the method. We use **bold** and underline to denote the best and second best.

| Learning-Based Method | Val14 | | Test-Hard | | Test-Random | |
|---|---|---|---|---|---|---|
| | NR | R | NR | R | NR | R |
| PDM-Open (Dauner et al., 2023) | 53.53 | 54.24 | 33.51 | 35.83 | 52.81 | 57.23 |
| UrbanDriver (Scheel et al., 2022) | 68.57 | 64.11 | 50.40 | 49.95 | 51.83 | 67.15 |
| PlanTF (Cheng et al., 2024b) | 84.72 | 76.95 | 69.70 | 61.61 | 85.62 | 79.58 |
| PLUTO (Cheng et al., 2024a) | 88.89 | 78.11 | 70.03 | 59.74 | 89.90 | 78.62 |
| DiffusionPlanner (Zheng et al., 2025) | 89.87 | 82.80 | 75.99 | 69.22 | 89.19 | 82.93 |
| SMART∗ (Wu et al., 2024) | 90.03 | 84.31 | **76.39** | 76.94 | 88.48 | 88.74 |
| Ours | **91.22** | **85.19** | 75.37 | **77.29** | **91.14** | **90.41** |

behavior under varying thresholds and correction lengths. In Section 5.3.4, we show the latency of our proposed method.

We further conduct additional experiments in the Appendix on 1) RL with the log-replay versus the learned world model in Section B.2.1, 2) zero-shot generalization in Section B.2.2, and 3) latency in Section 5.3.4.

### 5.3.1. EFFECTIVENESS OF THE CORRECTION TRACE

We compare our approach against an alternative self-correction method without the correction trace to show that the correction trace is important in correcting the behavior and reducing collisions. In CorrectionPlanner, we use a self-attention encoder to aggregate the full correction trace, and the policy generates a new token conditioned on this aggregated trace. We consider a simpler variant that does not use the trace and conditions only on the most recently proposed unsafe token, updating the proposal distribution for the next token. We show the results in Table 4. Our method achieves lower collisions with comparable progress, indicating that conditioning on the entire correction trace provides additional useful information beyond the most recent unsafe token, leading to more effective self-correction.

### 5.3.2. COMPARISON WITH DIFFERENT BASELINES

**Comparison with different *training* frameworks.** We compare our method with different training frameworks, with/without RL, and with/without self-correction. Specifically, we compare with 1) **IL**: imitation learning (IL) without RL and self-correction, 2) **IL (w/ corr.)**: imitation learn-ing without RL training but with self-correction, 3) **RL (w/o corr.)**: model trained with IL as well as pure RL, but without self-correction, and 4) **CorrectionPlanner (w/o corr.)**: our self-correction planner, but without self-correction during inference. As shown in Table 5, our method consistently outperforms the baselines in collision and achieves comparable progression. Comparing our method with IL (w/ corr.) and (w/o corr.), we show that IL itself is unable to perform effective self-correction. Also, comparing our method with RL (w/o corr.) and CorrectionPlanner (w/o corr.), we see that the policy improves with the RL training. However, the collision reduction is not solely from a stronger policy: CorrectionPlanner (w/o corr.) matches the pure RL baseline in both collision and progression, indicating that self-correction is able to avoid collision.

**Comparison with different *sampling* frameworks.** To further show that our model's self-correction is superior to other *sampling* frameworks and the self-correction cannot be achieved just through resampling, we compare with two standard sampling-based methods: **(1) Rejection Sampling.** When a proposed token is predicted unsafe, we *reject* it and resample from the policy's *original* token distribution (i.e., without utilizing the correction trace), **(2) Candidate Selection.** We sample 10 candidate ego trajectories from the policy, filter out colliding trajectories, and select the remaining trajectory with the highest progression rate. If all trajectories collide, we select the one with the greatest progression. Results are reported in Table 6.

As shown in Table 6, CorrectionPlanner outperforms both

*Table 3.* Planning performance comparison on Val14 NR.

| Type | Planner Score ↑ | Collisions ↓ | TTC ↑ | Drivable ↑ | Comfort ↑ | Progress ↑ | Speed ↑ |
|------|-----------------|--------------|-------|------------|-----------|------------|---------|
| SMART | 90.03 | 2.70 | 91.79 | 99.02 | 99.34 | 91.68 | 97.38 |
| Ours | **91.22** | **2.04** | 92.25 | 98.76 | 99.31 | 91.49 | 97.49 |

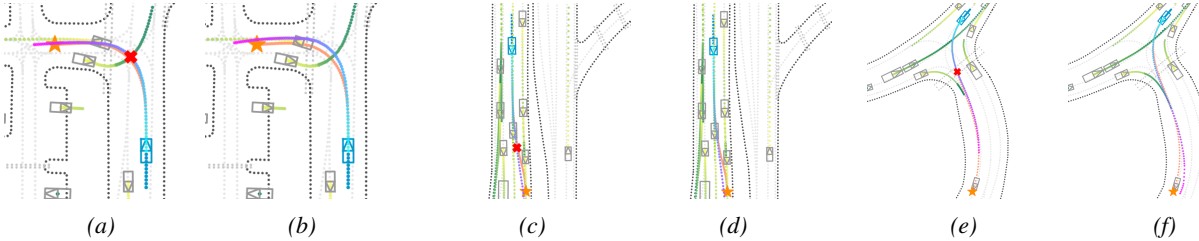

|     |     |     |     |     |     |
|-----|-----|-----|-----|-----|-----|
| *(a)* | *(b)* | *(c)* | *(d)* | *(e)* | *(f)* |

*Figure 4.* Visualization on self-correction. (a),(c) and (e): our method *w/o* self-correction in inference. (b), (d) and (f): our method *w/* self-correction. The orange trajectory is the log trajectory. Our method avoids the collision in these scenarios.

*Table 4.* Comparison with alternate self-correction baseline on WOMD.

| Method | Collision ↓ | Progression ↑ |
|--------|-------------|---------------|
| Alternation (w/o correction trace) | 2.01 | 0.9501 |
| Ours (w/ correction trace) | 1.68 | 0.9423 |

*Table 5.* Comparison across training frameworks.

| Method | Collision ↓ | Progression ↑ |
|--------|-------------|---------------|
| **Waymo** | | |
| IL | 2.34 | 95.77 |
| IL (w/ corr.) | 2.31 | 95.41 |
| RL (w/o corr.) | 2.21 | 94.98 |
| CorrectionPlanner (w/o corr.) | 2.20 | 95.02 |
| CorrectionPlanner | **1.68** | 94.23 |
| **nuPlan** | | |
| IL | 2.63 | 92.31 |
| IL (w/ corr.) | 2.58 | 92.18 |
| RL (w/o corr.) | 2.39 | 91.98 |
| CorrectionPlanner (w/o corr.) | 2.45 | 92.13 |
| CorrectionPlanner | **2.04** | 91.49 |

*Table 6.* Comparison with rejection resampling and candidate selection method.

| Setting | Method | Waymo | | nuPlan | |
|---------|--------|-------------|--------|-------------|--------|
| | | Collision↓ | Prog.↑ | Collision↓ | Prog.↑ |
| Rej. Sampling | IL | 2.31 | 95.76 | 2.57 | 92.11 |
| | RL | 2.09 | 94.18 | 2.39 | 91.98 |
| Candidate Sel. | IL | 2.29 | 94.99 | 2.51 | 91.48 |
| | RL | 2.14 | 94.21 | 2.37 | 91.33 |
| Self-Correction | Ours | **1.68** | 94.23 | **2.04** | 91.49 |

*Table 7.* Waymo collision critic statistics under different thresholds. CL denotes correction length. AvgTok. is the average number of correction tokens we generate per trajectory when self-correction occurs.

| /Threshold | 0.7 | | | 0.75 | | | 0.8 | | |
|------------|-----|---|---|------|---|---|-----|---|---|
| Precision | 0.21 | | | 0.68 | | | 0.87 | | |
| Recall | 0.93 | | | 0.74 | | | 0.31 | | |
| Time to Collision | 7.4 | | | 4.2 | | | 1.9 | | |
| | AvgTok | Coll. | Prog. | AvgTok | Coll. | Prog. | AvgTok | Coll. | Prog. |
| CL = 1 | 9.7 | 1.66 | 83.77 | 2.4 | 2.02 | 94.81 | 1.5 | 2.11 | 94.91 |
| CL = 2 | 11.4 | 1.49 | 80.32 | 4.2 | 1.85 | 94.48 | 2.9 | 2.06 | 94.75 |
| CL = 5 | 15.7 | 1.47 | 77.50 | 7.1 | **1.68** | 94.23 | 3.2 | 2.00 | 94.68 |

strategies. Rej. Sampling only avoids very limited collisions because repeated actions drawn from the same token distribution tend to remain in a similar region, i.e., close to the original unsafe token; so collisions still exist. In contrast, our self-correction generates new token conditions on the correction trace, which can shift the conditional distribution of subsequent proposals and escape such collision regions. Candidate selection can reduce collisions by selecting the best rollouts, but still underperforms our approach and achieves performance comparable to the pure RL baseline, probably because they share a similar spirit: "search and select".

### 5.3.3. RESULTS ON CLASSIFICATION THRESHOLD AND CORRECTION LENGTH

We study how the *collision classification threshold* and the *maximum correction length* (i.e., the maximum number of correction tokens in the correction trace per planning step; denoted as *correction length*) affect safety and progression.

**Precision/Recall of the Collision Critic.** Table 7 summarizes the statistics of the collision critic under different thresholds. Increasing the threshold introduces a precision-recall trade-off that directly affects when the correction module is triggered. At threshold $0.8$, precision increases but recall drops to $0.31$, meaning that about $70\%$ of potential collisions are missed, and the critic flags risk only $1.9$ (time to collision) planning steps ($\sim$1s) before collision, leaving little time for correction. And we see that a longer correction length does not improve the collision. On the other hand, at a threshold of $0.7$, it has a high recall and can detect over $90\%$ of future collisions, but it has a much lower precision, where around $80\%$ of predicted collisions are safe, and the first time it detects the collision is very early, triggering the self-correction more frequently and

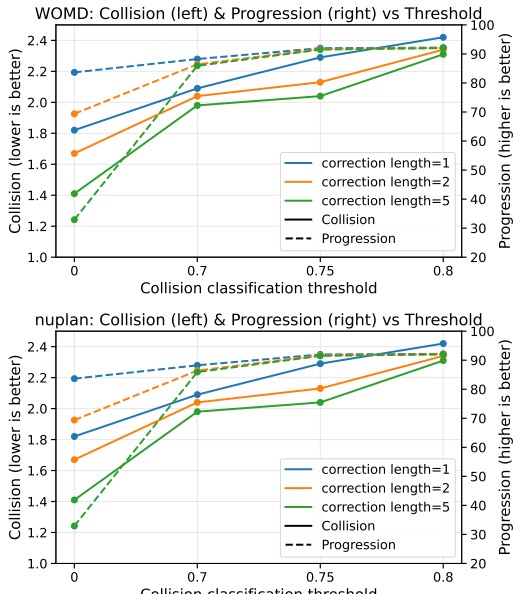

*Figure 5.* Effect of collision classification threshold and correction length on safety and progression.

much earlier, thus performing more self-correction steps. Though it reduces collisions, we see a significant loss of progress. On the other hand, at a threshold of $0.75$, we see a more balanced trade-off between precision and recall, as well as earlier detection of the collision. This leads to more reasonable self-correction traces, triggered timing, and the correction tokens we generate. Thus, we obtain a more balanced trade-off in collision and progression. We further conduct hyperparameter analysis of $k$ for the critic in Section B.2.3.

In Figure 5, we further visualize the progression and collision with different classification thresholds and correction lengths on two datasets. Similarly, we observe a clear trend that **a longer correction length reduces collisions**. This shows that our proposed method has self-correction ability with a larger correction budget.

Regarding the different thresholds, when we set the classification threshold to 0 (i.e., self-correction at every step), we observe a drastic decrease in progression, indicating that correction at every planning step is not feasible in terms of both performance and inference speed. And this also shows that our method doesn't reduce collisions by using a more complicated model that performs multiple forward passes with a higher-capacity model, but instead selectively performs self-correction on potentially unsafe actions. When we are more prone to self-correction with a lower threshold, we experience a lower collision rate but also a slightly lower progression rate. In both datasets, we observe that when the threshold $= 0.75$ and the correction length $= 5$, the progression-collision trade-off is optimal. We also tested $0.75$ as the threshold with a higher correction length, such as

8 or 10, but it does not improve performance further due to 1) the unavoidable not-at-fault collision and 2) the planner being stuck in the local unsafe region.

Overall, these results demonstrate that the model exhibits self-correction, with improved self-correction performance at larger self-correction budgets and more frequent triggering. Also, we show that self-correction is not achieved by simply running multiple forward passes of the policy, but rather by selectively applying self-correction via the correction trace on unsafe actions.

### 5.3.4. LATENCY

We report the inference latency on WOMD for different collision classification thresholds and the maximum reasoning length at each planning step in Table 8. SMART runs at $0.329s$ per trajectory, while our base policy without correction incurs a comparable latency of $0.357s$, indicating that our planner introduces negligible computational overhead. The best-performing hyperparameter set, with $0.75$ as the threshold and $5$ as the correction length, achieves $0.434s$ latency and doesn't significantly increase inference time, demonstrating that our method improves closed-loop safety and robustness *without significantly increasing latency*.

*Table 8.* Waymo latency (seconds) under different collision classification thresholds per trajectory. The best-performing hyperparameters, with 0.75 as the threshold and 5 as the correction length, don't significantly increase inference time. Experiment conducted on one H800.

| Method / Threshold | 0.00 | 0.70 | 0.75 | 0.80 |
| --- | --- | --- | --- | --- |
| SMART | 0.329 | – | – | – |
| SFT | 0.357 | – | – | – |
| CorrectionPlanner | | | | |
| Correction length = 1 | 0.917 | 0.497 | 0.392 | 0.387 |
| Correction length = 2 | 0.994 | 0.525 | 0.413 | 0.407 |
| Correction length = 5 | 1.162 | 1.036 | 0.434 | 0.431 |

## 6. Conclusion

In this paper, we propose a self-correction planner that explicitly corrects unsafe actions via a correction trace at each planning step. We train it using imitation learning, followed by RL, to achieve this self-correction capability. The empirical results show that our self-correction module effectively avoids collisions and exhibits diverse correction behaviors. We view this motion-token-level self-correction as a promising direction for bridging the gap between LLMs that perform reasoning through language. Future work includes considering a more calibrated collision critic and extending self-correction to other metrics beyond collision.

## Impact Statement

This paper proposes a self-corrective trajectory planner for the autonomous driving problem. Our approach could have positive impacts by reducing collision rates and improving the robustness of learned planners in complex multi-agent interactions through self-correction with a correction trace, potentially translating into safer driving behavior. And the zero-shot generalization ability makes it more scalable. The method is also general and could be applied to other robotics settings where safety is important.

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

# A. Algorithm

We summarize world model training, imitation learning, RL training, and inference in Algorithms 1 to 4.

---

**Algorithm 1** World Model Training

---

**Input:** Dataset $\mathcal{D}$, world model $p_\phi$, motion tokenizer $\mathcal{T}$, horizon $T$
**Output:** Trained world model $p_\phi$
**for** *each trajectory in the dataset* **do**
    Let the trajectory be $\tau \in \mathcal{D}$
    Tokenize all agent trajectories.
    Encode map and scene context $m$
    Compute next-token prediction loss:

$$\mathcal{L}_{\text{world}}(\phi) = -\sum_{t=1}^{T}\sum_{n=1}^{N} \log p_\phi\left(s_t^n \mid s_{<t}^{1:N}, m\right)$$

    Update $\phi$ by minimizing $\mathcal{L}_{\text{world}}(\phi)$
Freeze the trained world model $p_\phi$ for later RL rollouts

---

**Algorithm 2** Imitation Learning

---

**Input:** Expert dataset $\mathcal{D}$, policy $\pi_\theta$, maximum correction length $C$, horizon $T$
Initialize $\mathcal{L}_{\text{imit}} \leftarrow 0$ and $\mathcal{L}_{\text{corr}} \leftarrow 0$
**for** *each expert trajectory in the dataset* **do**
    Let the trajectory be $\tau \in \mathcal{D}$
    Tokenize ego and agent trajectories.
    **for** $t = 1, \ldots, T$ **do**
        Add next-token imitation loss:

$$\mathcal{L}_{\text{imit}} \leftarrow \mathcal{L}_{\text{imit}} - \log \pi_\theta\left(s_t^{\text{ego}} \mid s_{<t}^{\text{ego}}, s_{<t}^{\text{agents}}, \hat{s}_t^{\text{agents}}, m\right)$$

        Initialize correction trace $\mathcal{C}_t \leftarrow \emptyset$
        **for** $c = 1, \ldots, C$ **do**
            Sample proposed ego token:
$$\hat{s}_{t,c}^{\text{ego}} \sim \pi_\theta\left(\cdot \mid s_{<t}^{\text{ego}}, s_{<t}^{\text{agents}}, \hat{s}_t^{\text{agents}}, m, \mathcal{C}_t\right)$$

            **if** *Collision happens* **then**
                Append unsafe proposal to correction trace:

$$\mathcal{C}_t \leftarrow \mathcal{C}_t \cup \left\{\hat{s}_{t,c}^{\text{ego}}\right\}$$

            Add correction imitation loss:

$$\mathcal{L}_{\text{corr}} \leftarrow \mathcal{L}_{\text{corr}} - \log \pi_\theta\left(s_t^{\text{ego}} \mid s_{<t}^{\text{ego}}, s_{<t}^{\text{agents}}, \hat{s}_t^{\text{agents}}, m, \mathcal{C}_t\right)$$

Update $\theta$ by minimizing:
$$\mathcal{L}_{\text{pretrain}} = \mathcal{L}_{\text{imit}} + \mathcal{L}_{\text{corr}}$$

---

---

**Algorithm 3** Reinforcement Learning with Self-Correction

---

**Input:** Pretrained policy $\pi_\theta$, frozen imitation policy $\pi_{\mathrm{IL}}$, frozen world model $\hat{P}_\phi$, collision critic $f_\psi$, dataset $\mathcal{D}$, maximum correction length $C$, threshold $\eta$, horizon $T$, KL weight $\lambda$

**Output:** RL fine-tuned policy $\pi_\theta$

**for** *each training iteration* **do**

    Sample an initial scene from $\mathcal{D}$  Initialize rollout trajectory $\tau \leftarrow \emptyset$

    **for** $t = 1, \ldots, T$ **do**

        Initialize correction trace $\mathcal{C}_t \leftarrow \emptyset$

        **for** $c = 1, \ldots, C$ **do**

            Sample ego motion token:

$$\hat{s}_{t,c}^{\mathrm{ego}} \sim \pi_\theta\big(\cdot \mid s_{<t}^{\mathrm{ego}}, s_{\leq t}^{\mathrm{agents}}, m, \mathcal{C}_t\big)$$

            Estimate future collision probability:

$$p_{\mathrm{col}} \leftarrow f_\psi\big(s_{\leq t}, \hat{s}_{t,c}^{\mathrm{ego}}\big)$$

            **if** $p_{\mathrm{col}} < \eta$ **then**

                Accept token $s_t^{\mathrm{ego}} \leftarrow \hat{s}_{t,c}^{\mathrm{ego}}$  **break**

            **else**

                Append rejected token to correction trace:

$$\mathcal{C}_t \leftarrow \mathcal{C}_t \cup \{\hat{s}_{t,c}^{\mathrm{ego}}\}$$

        **if** *no token is accepted* **then**

            Execute final proposed token $s_t^{\mathrm{ego}} \leftarrow \hat{s}_{t,C}^{\mathrm{ego}}$

        Roll out next state with frozen world model:

$$s_{t+1} \sim \hat{P}_\phi\big(\cdot \mid s_t, s_t^{\mathrm{ego}}\big)$$

        Store executed token and correction trace in $\tau$

    Compute trajectory reward:

$$r(\tau) = \mathrm{Progression}(\tau) \cdot \mathbf{1}[\mathrm{Collision}(\tau) = 0] - \mathbf{1}[\mathrm{Collision}(\tau) = 1]$$

    Compute policy-gradient objective with KL regularization:

$$\mathcal{L}_{\mathrm{RL}}(\theta) = -\sum_{t=1}^{T} r(\tau) \log \pi_\theta\big(s_t^{\mathrm{ego}} \mid s_{<t}^{\mathrm{ego}}, s_{\leq t}^{\mathrm{agents}}, m, \mathcal{C}_t\big) + \lambda \mathrm{KL}\big(\pi_{\mathrm{IL}} \,\|\, \pi_\theta\big)$$

    Update $\theta$ by minimizing $\mathcal{L}_{\mathrm{RL}}(\theta)$

---

---

**Algorithm 4** Inference

---

**Input:** Policy $\pi_\theta$, collision critic $f_\psi$, maximum correction length $C$, threshold $\eta$, horizon $T$
**Output:** Executed ego motion tokens $s_{1:T}^{\text{ego}}$
Initialize scene history, map context $m$, and ego history $s_{<1}^{\text{ego}}$
**for** $t = 1, \ldots, T$ **do**
    Initialize correction trace $\mathcal{C}_t \leftarrow \emptyset$
    **for** $c = 1, \ldots, C$ **do**
        Sample or select ego motion token:
$$\hat{s}_{t,c}^{\text{ego}} \sim \pi_\theta\big(\cdot \mid s_{<t}^{\text{ego}}, s_{\leq t}^{\text{agents}}, m, \mathcal{C}_t\big)$$
        Estimate collision probability:
$$p_{\text{col}} \leftarrow f_\psi\big(s_{\leq t}, \hat{s}_{t,c}^{\text{ego}}\big)$$
        **if** $p_{\text{col}} < \eta$ **then**
            Accept token $s_t^{\text{ego}} \leftarrow \hat{s}_{t,c}^{\text{ego}}$ **break**
        **else**
            Append rejected token to correction trace:
$$\mathcal{C}_t \leftarrow \mathcal{C}_t \cup \{\hat{s}_{t,c}^{\text{ego}}\}$$
    **if** *no token is accepted* **then**
        Execute final proposed token $s_t^{\text{ego}} \leftarrow \hat{s}_{t,C}^{\text{ego}}$
    Execute $s_t^{\text{ego}}$ and update scene history

---

# B. Additional Experiments

## B.1. Dataset Details

We evaluate our method in reactive and non-reactive modes: reactive agents follow an IDM policy (Treiber et al., 2000), while non-reactive agents use log-replay trajectories. **WOMD**: Each scenario has a sequence length of 8 seconds and 1 seconds history, recorded at 10 Hz. Agents are controlled at 10 Hz, with a maximum of 128 per scenario. The training set has 487,002 scenarios, while the validation set has 44,096 scenarios. The simulation will not terminate until it reaches a maximum length of 8 seconds. We use the Waymax simulator (Xiao et al., 2025) to evaluate the closed-loop simulation. **nuPlan:** The nuPlan datasets contain over 1,300 hours of expert driving logs collected across four cities, covering diverse and challenging urban driving scenarios. Following the DiffusionPlanner (Zheng et al., 2025), we sample 1M training examples for training. Evaluation is performed on the standard Val14, Test14-random, and Test14-hard splits (Dauner et al., 2023).

## B.2. Additional Ablation studies

### B.2.1. LOG-REPLAY VERSUS LEARNED WORLD MODEL IN RL TRAINING

We further conduct an ablation in which RL fine-tuning is performed using log-replay only, i.e., without the frozen reactive world model, to show that using reactive modeling of the surrounding agents is beneficial to the RL training. The results are in Appendix B.2.1.

This ablation shows a clear pattern. Under the non-reactive evaluation setting, the log-replay-only variant achieves slightly lower collision than the world model variant. We believe this is expected, since both training and evaluation in this case use non-reactive agent behavior, so log replay is better matched to that evaluation protocol.

In contrast, under the reactive evaluation setting, the version trained with the reactive world model achieves better collision performance (1.68 vs. 1.83). This potentially suggests that reactive rollouts during RL provide a more suitable training signal when the test-time environment also contains interactive agent responses.

We emphasize that our motivation for using a reactive world model is not merely to improve performance under one benchmark setting, but to better capture the interactive nature of real-world driving, where surrounding agents respond to

the ego vehicle's behavior. From this perspective, we view the reactive evaluation setting as more realistic, and the above ablation supports our design choice of using a reactive world model in the main method.

At the same time, the log-replay-only variant still substantially outperforms SMART in both settings, indicating that the benefit of our method does not come solely from the world model itself; rather, the world model mainly helps when training in interactive, reactive environments.

*Table 9.* Comparison of our method with world model and log-replay training on WOMD under reactive and non-reactive evaluation settings.

| Method | Reactive | | | Non-Reactive | | |
|---|---|---|---|---|---|---|
| | Collision ↓ | Offroad ↓ | Progression ↑ | Collision ↓ | Offroad ↓ | Progression ↑ |
| SMART (Wu et al., 2024) | 2.36 | **0.87** | 91.33 | 4.30 | **0.86** | 90.87 |
| Ours (Log-replay for RL training) | 1.83 | 0.88 | **95.32** | **2.39** | 0.92 | **96.01** |
| Ours | **1.68** | 0.94 | 94.23 | 2.43 | 0.92 | 95.75 |

### B.2.2. ZERO-SHOT GENERALIZATION ABILITY

In Table 10, we show the zero-shot generalization ability of our method. We train the policy on the nuPlan dataset and evaluate it on the Waymax simulator. Specifically, we see that the performance of the Zero Shot (w/ corr.) improves the collision of Zero Shot (w/o corr.), demonstrating the effectiveness in *zero-shot self-correction generalization*. Also, compared with other baseline methods, Zero Shot (w/o corr.) shows low collision rates, demonstrating its zero-shot generalization ability. This zero-shot generalization might be due to the spatio-temporal attention of the agents, ego, and map, which is invariant to translation and rotation, thereby enabling good generalization.

*Table 10.* Zero-shot generalization ability of our method. Policy trained on the nuPlan dataset, evaluated on the Waymax simulator. The zero-shot (w/ corr.) is our method, and it achieves the second-lowest collision rate, alongside our method trained on WOMD, demonstrating zero-shot generalization and self-correction.

| Method | Collision ↓ | Offroad ↓ | Progression ↑ |
|---|---|---|---|
| EasyChauffeur-PPO (Xiao et al., 2024) | 4.71 | 3.81 | 97.93 |
| PlanT (Renz et al., 2022) | 2.94 | 1.65 | 95.85 |
| LatentDrive (Xiao et al., 2025) | 3.27 | 2.33 | 98.70 |
| SMART (Wu et al., 2024) | 2.36 | 0.87 | 91.33 |
| Zero Shot (w/o corr.) | 2.93 | 1.40 | 93.64 |
| Zero Shot (w/ corr.) | 2.62 | 1.26 | 92.83 |
| Ours | **1.68** | 0.94 | 94.23 |

### B.2.3. HYPERPARAMETER OF THE COLLISION CRITIC

*Table 11.* Experiment results on different $k$ when training the collision critic on WOMD.

| k | Collision ↓ | Progression ↑ |
|---|---|---|
| 1 | 2.19 | 96.53 |
| 3 | 1.97 | 94.99 |
| 5 | 1.68 | 94.23 |
| 7 | 1.63 | 91.42 |

In this section, we present how the hyperparameter of training the collision critic affects the self-correction ability. As shown, the objective of training the collision critic is:

$$\mathcal{L}_{\text{critic}}(\theta) = -\sum_{t=1}^{T} I(\text{collision}_{t:t+k} = 1) \log p_\theta\big(\text{collision} \mid s_{\leq t}\big) + I(\text{collision}_{t:t+k} = 0) \log p_\theta\big(\text{safe} \mid s_{\leq t}\big),$$

where $I(\text{collision}_{t:t+k} = 1)$ and $I(\text{collision}_{t:t+k} = 0)$ are collision or safe in $k$ future planning steps, respectively.

We conduct experiments with different values of $k$, which control when we trigger self-correction (i.e., how many steps ahead the collision predictor flags risk). We set $k \in 1, 3, 5, 7$. When $k = 1$, collisions are detected only when they happen. This leaves little time for the planner to correct, and we observe the highest collision rate for this hyperparameter. As $k$ increases, collisions decrease consistently, indicating that earlier warnings provide more time for effective correction. However, when $k = 7$, progression drops substantially while collision improves only marginally, suggesting that $k = 7$ might trigger self-correction too early and too often. As discussed in Appendix 5.3, overly frequent triggering can lead to conservative behavior, sacrificing progression without a commensurate gain in safety.

### B.3. Implementation Details

#### B.3.1. ARCHITECTURE

We use a discrete motion token vocabulary of size 1024 for the agent categories, including vehicle, pedestrian, and cyclist, where each token is a 0.5-second movement segment. The ego shares the same token as other vehicles.

**World model.** The world model is a 6-layer Transformer decoder, each consisting of 3 attention blocks with a hidden dimension of 128, and the attention head is 8. We use a two-layer MLP as the prediction head to map the hidden embedding to the token distribution.

**Policy.** Similarly, the policy model is a 2-layer Transformer decoder, each consisting of 4 attention blocks with a hidden dimension of 128, and the attention head is 8. We use a two-layer MLP as the prediction head to map the hidden embedding to the token distribution.

**Road Encoder.** The map and navigation information are both processed into road tokens, which contain three types of information: the position of each road token point, the road token direction at each point, and the type of each road token. An encoder transformer with 3 attention layers and 128 token embedding is applied to encode map tokens, and the attention head is also set to be 8.

**Critic Model.** The critic model is a 2-layer Transformer decoder, each consisting of 2 attention blocks with a hidden dimension of 128, and the attention head is 8. We use a two-layer MLP as the prediction head to map the hidden embedding to the binary output.

#### B.3.2. TRAINING

For data preprocessing, we set the maximum number of neighboring vehicles to 32 and the maximum number of lanes to 70. For navigation, we use 25 lanes for the nuPlan dataset and up to 5 lanes for the WOMD.

The world model and the imitation learning model are trained together for all three agent types using the AdamW optimizer (Loshchilov & Hutter, 2017). Both the dropout rate and the weight decay rate are set to 0.1. The learning rate is decayed from 0.0003 to 0 using a cosine annealing scheduler. We set the agent-agent cross-attention to a radius of 60, the map-agent to 30, and the map-map to 10. The training is set to be 16 epochs.

For RL training, the dropout and weight decay rates are set to 0.1. The learning rate is decayed from 0.0003 to 0 using a cosine annealing scheduler. The KL divergence weight is set to 0.1. The RL training is set to be 3 epochs. For the correction length per planning step, we randomly draw from $[0, 6]$, as 1) we want the model to have the ability to perform correction with a limited budget, and 2) some scenarios do not require too many self-correction budgets, and a larger budget might lead to an overfitting issue. We also apply batch normalization to the trajectory reward to stabilize training. For the collision critic, we set $k$ to 5 during the training. We further select approximately $100,000$ hard examples that may have collisions for RL training to improve training efficiency, where the hard example is the trajectories that may have collisions rollout from the policy and world model.

For the critic training, we use trajectory data generated by the policy and the world model during RL training. Specifically, we discard the correction trace in the data when training the critic, and only use the executed tokens. As the collision sample size is significantly smaller than the safe sample, we downsample the safe sample to a $1 : 1$ ratio to balance the training.

All experiments are conducted on 8 NVIDIA H800 GPUs.

### B.3.3. DETAILS OF TRAJECTORIES SAMPLING

**Rule-based correction samples.** For the RL training, positive reward samples are important to guide the policy towards a better performance, while negative reward samples will always make the sampled token probability become smaller, resulting in a more uniform distribution. In the self-correction training, one challenge we have is that the portion of the *corrected trajectories* is very small, thus most of the trajectories sampled are negative reward, making the training unstable and collapse as the imitation learning phase hasn't seen these samples and doesn't have enough ability to correct the unsafe action. One common solution is to use *rule-based samples*, i.e., to rollout trajectories by replaying the ground-truth for both agents and the ego.

# C. Visualization

In this section, we provide additional visualizations of the collision and self-correction trajectories.

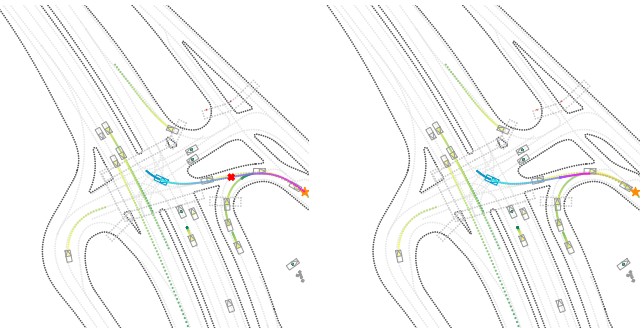

*Figure 6.* Slow down and yield to the right-turning vehicle.

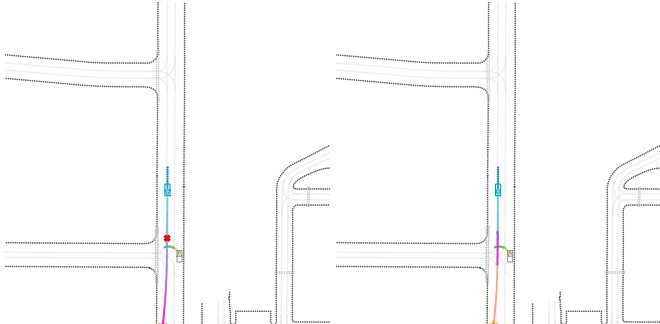

*Figure 7.* Slow down and yield to the right-turning vehicle, re-accelerate afterward.

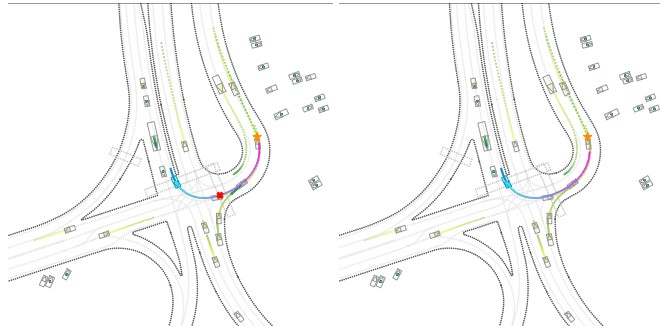

*Figure 8.* Adjust the speed in the left turn to avoid a collision.

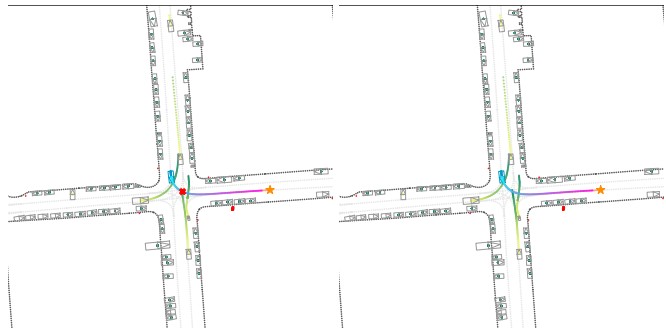

*Figure 9.* Adjust the left turn to avoid a collision in the intersection.

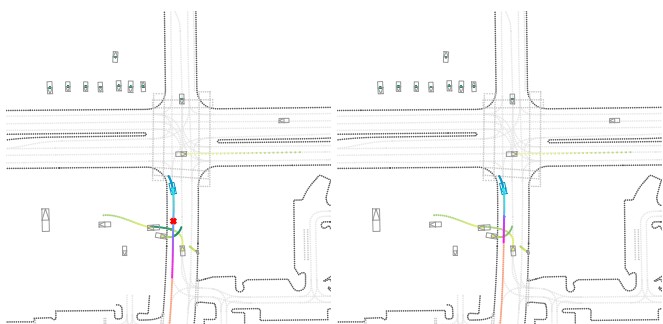

*Figure 10.* Adjust the speed and yield to multiple left-turn vehicles.

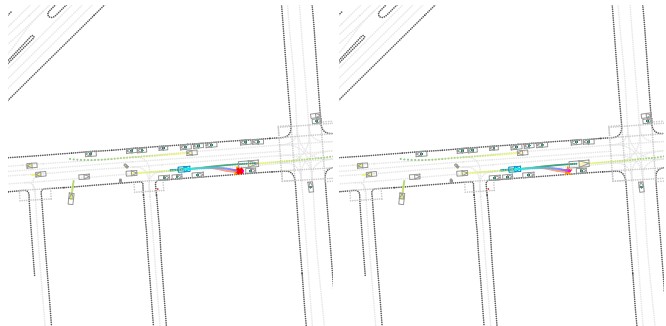

*Figure 11.* Brake in advance to avoid collision.

