# OpenReview forum: "CorrectionPlanner: Self-Correction Planner with Reinforcement Learning in Autonomous Driving"
_ICML.cc/2026/Conference — ICML 2026 regular_

### Official Review · Reviewer_6HED · 2026-02-26

**Soundness:** 3
**Presentation:** 3
**Significance:** 3
**Originality:** 3
**Overall Recommendation:** 4
**Confidence:** 4

**Summary:**

This paper proposes CorrectionPlanner, which formulates planning as the stepwise generation of discrete motion tokens and introduces a closed-loop propose–evaluate–correct self-correction mechanism within each planning step. The policy first proposes a candidate motion token. A collision critic then predicts whether executing this token would lead to a collision within a short time horizon. If deemed unsafe, the token is not executed; instead, it is appended to a historical correction trace, and the policy regenerates a new candidate conditioned on this trace. This process iterates until the safety criterion is satisfied or a maximum correction length is reached. The training procedure consists of two stages. First, imitation learning is used for next-token prediction, during which collision scenarios are explicitly exposed so that the model learns to perform corrections conditioned on failure traces. Second, reinforcement learning with a learned world model is applied to further optimize the policy, aiming to encourage earlier corrections and stronger safety performance. Experiments show more than a 20% reduction in collision rate on Waymax and near state-of-the-art planning scores on nuPlan.

**Compliance With Llm Reviewing Policy:**

Affirmed.

**Final Justification:**

The author's response was comprehensive and sufficient, and my concerns have been addressed. The core contribution of this paper remains technically sound. Therefore, I give a positive recommendation.

**Key Questions For Authors:**

Please see the weaknesses for questions.

**Limitations:**

yes

**Strengths And Weaknesses:**

### Strengths

1. Motion-token correction traces as structured signals analogous to Chain-of-Thought represent an insightful contribution. Compared to simple candidate selection or rejection sampling, the proposed method emphasizes that the correction trace alters the conditional distribution of subsequent token generation. This constitutes a stronger form of adaptive regeneration based on failure history.

2. The method design is clear and well aligned with the stated objectives. By rejecting unsafe tokens and encoding failure history into a correction trace that conditions subsequent generation, the approach elevates self-correction from post-hoc filtering to an explicit update of the policy distribution. The paper also carefully distinguishes this mechanism from simple rejection sampling.


### Weaknesses

1. The paper motivates the use of a reactive world model for rollouts to simulate multi-agent responses to the ego vehicle’s behavior, which better reflects interactive driving compared to non-reactive log-replay training. However, the experimental section does not provide an ablation study comparing RL rollouts using log-replay versus those using the learned world model. Including such an ablation would help substantiate the claimed benefits of reactive simulation.

2. The RL reward design only includes collision and progress signals, with the argument that incorporating too many reward terms may complicate training. However, many RL-based driving approaches include additional signals (e.g., off-road penalties or time-to-collision constraints). Is there empirical evidence demonstrating that additional reward components indeed complicate optimization in this setting? Intuitively, incorporating richer safety-related signals could be beneficial.

3. Providing more algorithmic pseudocode would substantially improve clarity and reproducibility. In addition, Figure 1, as a teaser figure, should more explicitly highlight the limitations of prior methods and clearly contrast them with the proposed approach, rather than merely listing three modeling paradigms.

---

> ### Author Rebuttal · Authors · 2026-03-31
>
> We thank the reviewer for the positive feedback. We now answer the reviewer's questions.
>
> ## Q1. Ablation on using log-replay during the RL training instead of the world model.
>
> ## A1.
>
> Thank you for this helpful suggestion. We have added an ablation in which RL fine-tuning is performed using log-replay only, i.e., without the frozen reactive world model.
>
>
> | Method | Reactive Collision ↓ | Reactive Offroad ↓ | Reactive Progression ↑ | Non-Reactive Collision ↓ | Non-Reactive Offroad ↓ | Non-Reactive Progression ↑ |
> |---|---:|---:|---:|---:|---:|---:|
> | SMART| 2.36 | 0.87 | 91.33 | 4.30 | 0.86 | 90.87 |
> |Ours Log-replay| 1.83 | 0.88 | 95.32 |  2.39 | 0.92 | 96.01
> | Ours | 1.68 | 0.94 | 94.23 | 2.43 | 0.92 | 95.75 |
>
> This ablation shows a clear pattern. Under the non-reactive evaluation setting, the log-replay-only variant achieves slightly lower collision than the world model variant. We believe this is expected, since both training and evaluation in this case use non-reactive agent behavior, so log replay is better matched to that evaluation protocol.
>
> In contrast, under the reactive evaluation setting, the version trained with the reactive world model achieves better collision performance (1.68 vs. 1.83). This potentially suggests that reactive rollouts during RL provide a more suitable training signal when the test-time environment also contains interactive agent responses.
>
> We emphasize that our motivation for using a reactive world model is not merely to improve performance under one benchmark setting, but to better capture the interactive nature of real-world driving, where surrounding agents respond to the ego vehicle’s behavior. From this perspective, we view the reactive evaluation setting as more realistic, and the above ablation supports our design choice of using a reactive world model in the main method.
>
> At the same time, the log-replay-only variant still substantially outperforms SMART in both settings, which also indicates that the benefit of our method does not come solely from the world model itself; rather, the world model mainly helps when training for interactive, reactive environments.
>
>
> ## Q2. The RL reward design only includes collision and progress signals.
>
> ## A2.
>
> Our intention of only including the collision and progress is that we are not trying to improve the performance using RL fine-tuning with a complicated reward design that matches the overall goal, like planning score in Nuplan. So we choose to use collision and progress as the reward. But we also agree that including more reward signals will improve overall performance.
>
>
> ## Q3. Pseudocode can improve clarity and reproducibility. Highlight the limitations of prior methods in Figure 1.
>
> ## A3.
>
> Thanks for pointing this out. We will include a pseudocode of two stage training and inference in our revision. We will also highlight the main difference and advantage of our method in Figure 1: compared with the vanilla autoregressive planner, our method has the ability to self-correct and reduce collisions, and compared with reasoning with language, our method performs correction through motion tokens rather than language reasoning.

---

> > ### Author Rebuttal · Reviewer_6HED · 2026-04-03
> >
> > Thank you for providing a comprehensive and detailed rebuttal. My concern has been addressed. Therefore, I maintain my positive recommendation.

---

> > > ### Author Response · Authors · 2026-04-07
> > >
> > > Thank you very much for your positive feedback! We will revise our paper according to your constructive reviews.
> > >
> > > Best,
> > >
> > > Authors

---

### Official Review · Reviewer_d8Av · 2026-03-11

**Soundness:** 3
**Presentation:** 3
**Significance:** 3
**Originality:** 3
**Overall Recommendation:** 4
**Confidence:** 3

**Summary:**

The paper presents CorrectionPlanner, an autonomous driving framework that incorporates a self-correction mechanism directly into the motion-token generation process. Inspired by CoT in LLMs, the authors implement a propose-evaluate-correct cycle. If the system predicts a trajectory (token) that the learned critic classifies as unsafe, this unsafe trajectory is retained in the context to help the model generate a safer alternative.  By treating past mistakes as informative correction traces rather than just discarding them, the model achieves a notable 20% reduction in collisions on Waymax and nuPlan. It’s a promising way to bridge the gap between autoregressive prediction and safety-critical planning.

**Compliance With Llm Reviewing Policy:**

Affirmed.

**Final Justification:**

I maintain my evaluation, and I remain concerned about the reliance on the critic and the potential overhead introduced by such a heavy world model.

**Key Questions For Authors:**

On average, how many correction iterations does the model perform per planning step? If it reaches the maximum length without finding a safe token, is there a hard-coded fallback like an emergency brake?

Without specific indicator tokens, how does the model internally distinguish between actual historical context and these rejected correction traces?

Can you provide the specific running fps compared to the SMART baseline to better quantify the computational overhead?

How does the world model handle rare "long-tail" events? I'm concerned that if the diversity of the learned correction strategies is limited by the world model's ability to simulate extreme scenarios.

**Limitations:**

yes

**Strengths And Weaknesses:**

**Strengths**

The adaptation of LLM-style self-correction to the motion planning domain is highly intuitive. Using rejected tokens as "reasoning traces" to guide the next attempt is a creative move.

The method avoids the black-box nature of many end-to-end planners by making the correction process explicit and integrated into the autoregressive generation.

The empirical gains are impressive, particularly the significant drop in collision rates on established benchmarks like nuPlan and Waymax.

**Weaknesses**

There is a clear concern regarding inference latency. An iterative generate-and-check loop adds computational weight, which might be a bottleneck for real-time deployment in high-speed or dense urban environments.

The system’s safety is entirely dependent on the critic’s accuracy. If the collision critic misses a hazard, the correction mechanism never triggers, potentially leading to failures. In addition, the model might become overly conservative to satisfy a very strict critic.

The reliance on a frozen world model for training faces a sim-to-real gap; if the world model has inherent biases in agent behavior, the learned correction strategies might not generalize well to real-world edge cases.

---

> ### Author Rebuttal · Authors · 2026-03-31
>
> We thank the reviewer for acknowledging the contribution of the paper and the experimental results. We now answer the reviewer's questions.
>
> ## Q1. The generate-and-check loop can be a bottleneck for real-time deployment in high-speed or dense urban environments.
> We report latency in Appendix A.2.3. Under our best hyperparameters (threshold = 0.75, correction length = 5), our method requires 0.434s per trajectory (8s horizon, 16 planning steps in total), compared with 0.329s for SMART and 0.357s for imitation-learning-only (without self-correction). Self-correction increases inference time, but we believe the overhead is still reasonable given the safety gains.
>
> ## Q2. The system’s safety depends heavily on the critic: missed hazards prevent correction, while an overly strict critic may make the planner too conservative.
> Thank you for raising this important concern. We agree that the critic is a key component: false negatives can miss hazards, while false positives can make the planner overly conservative. However, the planner is not entirely dependent on the critic for safety, and the critic serves as an additional correction mechanism on top of the base planner, rather than the only control of safe behavior.
>
> We study the trade-off of the precision/recall in Table 6. At a low threshold, the critic achieves high recall (e.g., 0.93) but loss progression, indicating more conservative behavior. At a threshold of 0.75, the planner is less conservative while still maintaining substantial recall and lower collision than the w/o baseline. This shows that the safety–efficiency trade-off induced by the critic is controllable.
>
> Moreover, the qualitative results in Figure 4 show that self-correction does not improve safety by being conservative: the planner may yield, adjust turning paths, or accelerate earlier depending on the scenario. Thus, the critic does not simply enforce a conservative fallback policy, but can trigger more flexible corrections.
>
> Overall, our results suggest that (1) the planner is not solely reliant on the critic for basic competence, (2) the critic-induced trade-off can be controlled through the threshold, and (3) the resulting corrections are not limited to conservative actions.
>
> ## Q3. The reliance on the world model for training faces a sim-to-real gap.
> Thank you for raising this important concern. We agree that using the world model introduces a potential sim-to-real gap, and this is a common limitation of recent autonomous driving RL methods such as CarPlanner and Plan-R1.
>
> However, a reactive world model trained on the same data remains important in our setting because self-correction can change surrounding agents’ future behavior, which non-reactive or log-replay rollouts cannot capture. To reduce over-reliance on the simulator, we also include log-replay-based training/ablation(Table in the response to Reviewer 6HED), showing that the model can still achieve self-correction and competitive results using log-replay for RL training.
>
> Moreover, Appendix A.2.2 shows zero-shot generalization to unseen datasets/settings. Although this does not fully resolve the sim-to-real gap, it suggests that our method is not merely overfitting to the world model's biases.
>
> ## Q4. How many correction iterations does the model perform per planning step? Any hard-coded fallback?
>
> As reported in Table 6, we set the correction iteration of each planning step to 5, and the average correction iteration of the entire trajectory is 4.2. For fair comparison with learning-based motion planning, we do not use any hard-coded fallback, and we are just a pure learning-based method.
>
>
> ## Q5. How does the model internally distinguish between actual historical context and these rejected correction traces?
>
> As shown in Figure 3(b), we include an additional encoder to encode the correction trace, enabling the model to distinguish it from the historical context.
>
>
> ## Q6. How does the world model handle rare "long-tail" events? Is the diversity of the learned correction strategies limited by the world model's ability to simulate extreme scenarios?
> We agree that long-tail events are a central challenge in autonomous driving. Since our world model is trained on Waymo/nuPlan, its ability to simulate extremely rare events is naturally limited by dataset coverage, and we do not claim it fully captures all long-tail scenarios.
>
> In our method, the world model is used to provide reactive multi-agent rollouts for RL for the nuPlan/Waymo data, not to enumerate all rare edge cases. The main contribution is the self-correction mechanism. We agree that under-representing rare events can limit the diversity of learned correction behaviors.
>
> This is an important future direction: a stronger world model trained on larger, more diverse data, or augmented with targeted rare-event simulation, would likely improve both RL robustness and correction diversity.

---

> > ### Author Rebuttal · Reviewer_d8Av · 2026-04-03
> >
> > Thanks for your detailed reply! Most of concerns have been addressed. The reliance on the critic and world model still makes me feel a bit overwhelmed, which reduces the ease of use of the method. Overall, I'm still maintaining my original score of 4 points.

---

> > > ### Author Response · Authors · 2026-04-07
> > >
> > > Thank you very much for your positive feedback! We will revise our paper according to your constructive reviews.
> > >
> > > Best,
> > >
> > > Authors

---

### Official Review · Reviewer_PhBq · 2026-03-12

**Soundness:** 2
**Presentation:** 3
**Significance:** 3
**Originality:** 3
**Overall Recommendation:** 4
**Confidence:** 3

**Summary:**

This paper proposes CorrectionPlanner, an autoregressive motion-token planner for autonomous driving with an explicit propose–evaluate–correct loop. At each planning step, the policy proposes a motion token, a learned collision critic predicts whether executing it would induce a near-future collision, and if the token is judged unsafe, the model appends it to a correction trace and generates a revised token conditioned on that trace. The paper presents this trace as a motion-token-space analogue of self-correction or reasoning traces in language models. Training is conducted in two stages: imitation learning via next-token prediction, followed by model-based reinforcement learning using rollouts from a frozen world model with reactive traffic agents. The paper reports more than 20% collision reduction on Waymax and improved planning scores on nuPlan, and includes ablations on RL, self-correction, correction length, and collision thresholds.

**Compliance With Llm Reviewing Policy:**

Affirmed.

**Final Justification:**

The authors address some of my concerns and improve my confidence in the paper, especially by clarifying that the gains are not simply due to reward design or a more conservative policy. My main concern is not fully resolved, since it is still somewhat unclear how much of the improvement is specific to the proposed self-correction mechanism itself rather than the broader combination of RL, critic guidance, and iterative replanning. Overall, the paper is strengthened and I am more positive about it, although I still retain some reservations.I can raise my score to 4, but my overall view remains somewhere between 3 and 4.

**Key Questions For Authors:**

1. Can the authors provide stronger evidence that the gains come specifically from the correction-trace mechanism, rather than more generally from RL fine-tuning plus a collision critic?

2. How reliable and calibrated is the learned collision critic, and how do its errors affect planning performance?

3. How much of the reported improvement depends on the particular frozen world model used during RL training?

4. Can the authors clarify more sharply what is fundamentally new in CorrectionPlanner relative to recent autoregressive or motion-token RL planners, such as CarPlanner and Plan-R1, beyond combining tokenized planning, safety evaluation, and RL fine-tuning?

5. What evidence supports the interpretation of the correction trace as an analogue of a reasoning trace, rather than simply a useful memory of rejected samples?

**Strengths And Weaknesses:**

This paper studies an important problem in autonomous driving planning: how to revise unsafe action proposals before execution rather than relying entirely on one-shot generation. The core intuition is reasonable and interesting. Instead of treating planning as a single-pass prediction problem, the method explicitly identifies unsafe motion-token proposals with a learned collision critic and conditions later proposals on a correction trace formed by previously rejected tokens. This gives the approach a more distinctive procedural identity than a standard RL fine-tuning pipeline. The paper also has a practical empirical focus, with closed-loop evaluation on both Waymax and nuPlan. The overall training setup—imitation learning followed by model-based reinforcement learning with a frozen world model—is consistent with current practice in learning-based planning. In addition, the ablations in Table 4 are useful in showing that the full method reduces collisions relative to IL-only, RL-without-correction, and correction-disabled variants.

That said, I have several important concerns across soundness, presentation, significance, and originality.

From a soundness perspective, the method is plausible, but the central claim is not yet supported strongly enough. The main issue is that the paper does not cleanly establish that the gains come from the proposed self-correction mechanism itself, rather than from a combination of a stronger policy, a learned collision critic, and RL fine-tuning. Table 4 is directionally useful, but it remains primarily an internal ablation across variants of the same framework, so it does not by itself fully establish that the correction trace—rather than the broader combination of RL fine-tuning, critic-based safety evaluation, and iterative replanning—is the decisive source of improvement. While the paper does include useful comparisons to rejection sampling, candidate selection, and a simpler alternate self-correction baseline, these comparisons are still not sufficient to fully isolate the unique contribution of the correction-trace mechanism relative to other plausible generate–evaluate–resample or generate-and-select alternatives. That matters because recent planning work already combines autoregressive planning with RL, tokenization, and world-model-based training. More broadly, recent work has already explored nearby combinations of autoregressive planning, tokenized motion generation, and RL-based refinement or alignment, so the paper would benefit from sharper positioning of what is fundamentally new in the correction-trace mechanism itself. The paper therefore needs stronger evidence that the correction trace is the decisive source of improvement, not just one reasonable implementation of a broader generate–evaluate–resample idea.

A second soundness concern is the collision critic. The entire correction loop depends on it, but the paper does not yet provide enough evidence that it is calibrated, robust, or more reliable than simpler alternatives. The discussion section itself acknowledges that a more calibrated collision critic is future work. This is a serious limitation because if the critic is the gatekeeper for all correction behavior, then the claimed self-correction ability is only as good as this binary safety oracle. The current paper provides some useful analysis of threshold sensitivity and precision–recall trade-offs, but still does not provide enough evidence about how critic false positives and false negatives affect planner behavior across scenario types, or whether the critic is robust enough to serve as a reliable gatekeeper for correction. Figure 5 and the associated discussion do show a trade-off between threshold choice, progression, and collision rate, which is helpful, but this is not sufficient validation of the critic as a reliable planning component.

A third soundness issue is the reward design and training story. The RL reward is extremely simple—progress if no collision, minus one for collision—and the paper explicitly says it avoids using more reward signals to simplify training.

That simplicity is understandable, but it raises the question of whether the observed gains are primarily due to stronger collision aversion under this particular reward shaping rather than a general self-correction capability. In addition, the paper relies on a pretrained frozen reactive world model during RL training, but it does not yet provide much evidence about how sensitive the learned policy is to the fidelity, bias, or design choices of that simulator. Since the world model is central to RL training, stronger validation of transfer from training dynamics to benchmark dynamics would improve confidence.

From a presentation perspective, the paper is generally readable and the main idea is easy to follow. The propose–evaluate–correct loop is clearly described, and the figures communicate the intended intuition well. The terminology of “correction trace” is memorable, and the paper’s narrative from one-shot autoregressive generation to explicit corrective planning is coherent. However, some of the framing is overstated. The analogy between motion-token correction traces and language-model reasoning traces is suggestive, but currently more rhetorical than technically substantiated. The paper has not shown that the trace has the interpretive or causal role that language reasoning traces are often claimed to have; it has shown that retaining rejected unsafe tokens can help condition later samples. Those are not the same thing. I would recommend more restrained wording here. The related-work positioning could also be sharper, especially in distinguishing the proposed correction-trace mechanism from nearby autoregressive, motion-token, and RL-based planning frameworks. The paper cites recent driving and RL planners, but it does not distinguish itself sharply enough from nearby autoregressive/RL/planning-as-token-generation work, especially given how quickly this area is moving.

From a significance perspective, the paper works on an important problem and does report meaningful closed-loop improvements. Reducing collision through an explicit corrective mechanism is a worthwhile direction, and if the method is robust, others may build on the idea of iterative corrective planning in token space. However, the broader significance is still moderate rather than strong. The contribution is primarily an architectural and training refinement within a fast-moving line of autonomous driving planners, rather than a broadly general ML contribution. For ICML, empirical planning papers usually need either especially strong generality, especially strong analysis, or a more clearly differentiated modeling principle. At present, this paper does not yet fully meet that bar.

From an originality perspective, the paper does contain a real idea: preserving a history of rejected unsafe motion tokens as a conditioning trace and using that trace inside an autoregressive correction loop. That is not a trivial restatement of standard policy improvement. However, the surrounding ingredients—motion-token autoregression, model-based RL, rule/critic-guided safety evaluation, world-model rollouts—are all already active areas of recent work. So the originality is meaningful but limited. The paper’s novelty lies more in the specific self-correction trace mechanism than in the overall planning framework, and the paper would be stronger if it isolated and justified that novelty more rigorously.

Overall, this is a promising and technically competent paper with a clear motivation and a genuinely interesting corrective-planning idea, but the current evidence is not yet strong enough for the level of claim the paper makes. Its main strengths are the practical relevance of the problem, the clear propose–evaluate–correct formulation, and the closed-loop gains on Waymax and nuPlan. Its main weaknesses are the incomplete isolation of the correction-trace mechanism, limited validation of the collision critic, insufficient comparison to alternative correction strategies, and originality that is meaningful but still somewhat narrow for ICML.

---

> ### Author Rebuttal · Authors · 2026-03-31
>
> We thank the reviewer for the positive feedback and questions. We address the concerns below.
> ## Q1. Can the authors provide stronger evidence that the gains come specifically from the correction trace, rather than from RL fine-tuning with a collision critic more generally?
> In Appendix A.2.1, we include an ablation in which iterative replanning conditions only on the most recently rejected unsafe token, rather than on the full correction trace. It performs worse than the full correction-trace version and is only slightly better than Rej. sampling. This suggests that the gain does not come simply from repeated replanning or a stronger policy (multiple forward of the same policy); the full correction trace provides additional information that helps reduce collisions. Together with the other ablations, this supports that the correction trace is a useful mechanism beyond RL fine-tuning, critic-based filtering, or simple generate-evaluate-resample loops. We believe that conditioning on the full correction trace might help the policy avoid repeatedly proposing similar unsafe solutions, whereas iterative replanning without this history might stay trapped in the same unsafe region.
> ## Q2. How robust is the critic, and is it well-calibrated, given that self-correction relies on it?
> We compare predicted collision probabilities with empirical collision frequencies in each probability bin on Waymo. The 1-step critic (predicts collision one planning step before the collision) is reasonably calibrated overall, though overconfident in the high-risk area; the 5-step critic (predicts collision five planning steps before the collision) is more overconfident, which is normal since longer-horizon future collision prediction is more uncertain. But importantly, this means it tends to flag risk rather than miss hazards.
>
> Table 6 further shows that, although the critic is not perfectly calibrated, it is still reliable enough to provide effective correction signals. We also evaluate robustness in Appendix A2.2 via zero-shot transfer to an unseen dataset, suggesting the generalization/robustness of our method. Overall, our results show that the critic is reasonably calibrated at short horizons, more over-confident at longer horizons, and effective as a gatekeeper in our planner with generalization ability.
> |Bin|1-step,Pred. Avg. Prob.|1-step, Real Collision Prob.|5-step Pred. Avg. Prob.|5-step,Real Collision Prob.|
> |-|-|-|-|-|
> |<0.1|0.06|0.01|0.07|0.01|
> |0.1–0.2|0.17|0.04|0.17|0.02|
> |0.2–0.3|0.24|0.12|0.26|0.05|
> |0.3–0.4|0.37|0.24|0.35|0.19|
> |0.4–0.5|0.46|0.41|0.47|0.29|
> |0.5–0.6|0.57|0.51|0.54|0.40|
> |0.6–0.7|0.64|0.55|0.66|0.52|
> |0.7–0.8|0.74|0.66|0.76|0.61|
> |0.8–0.9|0.83|0.77|0.87|0.72|
> |>0.9|0.97|0.96|0.96|0.86|
>
> ## Q3. The reward is simple. Does self-correction mainly come from strong collision aversion?
> The gains are not mainly due to reward design. The pure RL baseline is trained with the same reward, yet still underperforms ours. So the improvement cannot be explained solely by stronger collision aversion due to the reward. Moreover, the qualitative results in Figure 4 show that self-correction does not simply make the policy more conservative: it generates different trajectory revisions such as adjusting turn shape, delaying lateral motion, or changing path to avoid collision compared with the pure RL method.
>
> ## Q4. How sensitive is the learned policy to the fidelity, bias, or design of the world model? How much of the reported improvement depends on the world model?
> Our method outperforms SMART, the world model used for training, suggesting that the policy can still learn effective self-correction under the imperfect or biased world model, rather than inheriting or being severely limited by its behavior. More broadly, a learned reactive world model is also a standard choice in RL training, as in CarPlanner and Plan-R1. We agree that it could be sensitive, and a strong/robust world model would likely further improve results.
>
> Our log-replay-only RL ablation (Table in the response to Reviewer 6HED) shows that even with log replay, our self-correction method still reduces collisions and obtains the self-correction ability, suggesting that the self-correction ability does not rely solely on the frozen world model, although the reactive world model provides further gains.
>
> ## Q5. Can you clarify the novelty relative to recent works? Are you overstating the correction trace as reasoning?
> Unlike prior RL-based works that improve performance through complex reward design, to the best of our knowledge, this is the first work to introduce an explicit correction trace with a propose–evaluate–correct mechanism. So the core novelty lies in the correction mechanism itself, achieved via RL, rather than in different styles of RL tuning.
>
> We use reasoning trace only as an analogy to explain the intuition, but do not intend to claim that they are the same. We will make it more precise and avoid overstating it.

---

> > ### Author Rebuttal · Reviewer_PhBq · 2026-04-03
> >
> > Thank you to the authors for the careful rebuttal. I appreciate the clarifications regarding the intended role of the correction trace, the collision critic, the world-model-based RL setup, and the reasoning-trace analogy. The responses help clarify the paper’s intended claims and make the overall positioning more precise.
> >
> > That said, my main concerns remain only partially resolved. First, I still do not think the current evidence fully establishes that the reported gains come specifically from the full correction-trace mechanism itself, rather than more broadly from the combined effect of RL fine-tuning, critic-guided safety filtering, and iterative replanning. The comparison to the recent-token-only variant is useful, but it still does not completely isolate the correction trace from these other interacting components. Second, while the critic analysis is helpful, it also confirms that the critic is imperfectly calibrated, especially at longer horizons, so I remain somewhat unconvinced that it has been validated strongly enough as the key gatekeeper for the correction loop. Finally, the clarification that the reasoning-trace language is only an analogy is appropriate, but it also reinforces my view that the main novelty lies in the specific correction-trace mechanism, rather than in a more broadly differentiated overall planning framework.
> >
> > Overall, the rebuttal improves the paper and addresses several important points, but the core concerns are, in my view, only partially resolved.

---

> > > ### Author Response · Authors · 2026-04-05
> > >
> > > We thank the reviewer again for all the great comments and constructive feedback, which improve the presentation and soundness, and we will incorporate them into our revision.
> > >
> > > ---
> > >
> > > ## Q1. It is unclear whether the gains stem from the correction-trace mechanism or from the combined effects of RL fine-tuning, safety filtering, and iterative planning.
> > >
> > > 1. **Our main contribution is the self-correction, which enables the planner to revisit proposed actions before execution, rather than the correction trace in isolation.** Specifically, the critic triggers correction, the correction trace provides information for an alternative action, and RL serves as the training mechanism that enables the model to learn this correction behavior. Although the correction trace is important and effective, we do not claim that it is the only contribution or the only driver of collision improvement. To our knowledge, such a self-correction design has not been studied in autonomous driving, and it cannot be achieved by safety filtering, iterative planning, or RL fine-tuning.
> > >
> > > 2. **The gain does not come from each component alone, or a simple additive combination of the critic, correction trace, replanning, and RL, where each component independently contributes a small collision reduction, which leads to the final improvement. Instead, they work together as an integrated system to reduce collisions.** Our ablations show that: *for each component itself, the collision reduction is only marginal or negligible*. Specifically, the critic alone can provide signals for rejection sampling or iterative planning, yet these variants yield marginal or no improvement, showing that merely detecting unsafe behaviors is insufficient for self-correction without additional mechanism design. Conversely, while the correction trace can support correction, it still requires a trigger. **The significant improvement appears only when these components are integrated into the self-correction loop**, in which the critic triggers the correction, and the correction trace enables meaningful revision. So, the two components function jointly as a system, rather than contributing independently in similar ways.
> > >
> > > 3. **RL is not only a fine-tuning step.** As shown in Table 4, IL with self-correction achieves performance comparable to IL-only but is much worse than CorrectionPlanner, suggesting that the self-correction ability is not learned during the IL. In contrast, RL enables the model to have this ability. So, RL is not just a fine-tuning component like Plan-R1, but a necessary training component for learning the self-correction behavior, as we sample rollout collision data with correction trace for training. And the RL itself, or the RL + recent-token-replanning ablation in Appendix A.2.1, is less effective as a self-correction.
> > >
> > > 4. **In summary, the improvement is not a simple combination of independently beneficial components.** Instead, **the components play distinct and complementary roles within the self-correction mechanism**: the critic provides the correction signal, the correction trace supports revision, and RL enables the model to learn self-correction with the correction trace. We will also revise the writing to ensure that the analogy does not give the wrong impression.
> > >
> > >
> > > ## Q2 Critic is imperfectly calibrated, is it enough as the key gatekeeper for the correction?
> > >
> > > **The critic can identify a large portion of collisions and can provide a meaningful safety signal for the correction loop, which is also very important comparing with perfect calibration.** This is supported by the precision–recall results and the observed collision reduction. Our calibration analysis also suggests that the critic is somewhat conservative: it is more likely to overpredict collision risk than to miss collisions, which is acceptable as a gatekeeper. Importantly, this conservatism is controllable. The precision–recall analysis shows that, by choosing different thresholds, we can obtain a reasonable trade-off between progress and collision. In practice, the critic captures around 70% of collision cases while remaining neither overly conservative nor overly aggressive, indicating that even an imperfectly calibrated critic can still provide useful and controllable signals for self-correction. **Therefore, given its acceptable precision–recall trade-off, reasonable calibration error, tunable threshold, and not directly using the probability,  the critic is valid for the self-correction.**
> > >
> > > **Ultimately, the overall safety performance still depends on the planner’s ability to generate safer behaviors and perform effective self-correction, rather than on the critic alone, and a perfect critic itself cannot improve the performance without a well-performing self-correction.** Given that a large portion of collisions can be identified, the main challenge is how to effectively use these safety signals to guide correction and planning while maintaining overall planning quality.

---

### Official Review · Reviewer_MLyu · 2026-03-13

**Soundness:** 3
**Presentation:** 3
**Significance:** 1
**Originality:** 1
**Overall Recommendation:** 2
**Confidence:** 4

**Summary:**

This paper proposes an actor-critic framework for safer autonomous driving.The key idea is to learn to adjust proposed (potentially unsafe) actions. The method is instantiated as first imitation learning, then model-based RL. It outperforms two open source baselines.

**Compliance With Llm Reviewing Policy:**

Affirmed.

**Final Justification:**

I will trust the editor to weight the other reviewers' assessment of novelty and relevance to the ICML community. The rebuttal somewhat assessed my concerns, but basically just confirms my stance: I think any safety-focused method has to compare against a baseline with a formal guarantee.

This is especially true because the method relies on generating future state (position/velocity) predictions of ego and other agent trajectories, which is exactly the data format that most safety-focused methods would need. So I disagree with the argument that this method is somehow fundamentally different and therefore can't be compared against existing safety methods.

Finally, I have a philosophical objection to this style of paper. Making an autonomous driving system a bit safer, but still unsafe, means that we know *before deployment* that the robot is going to crash into people if it is deployed. Instead, I believe that such a problem setting *requires* us to make strict guarantees before any real-world deployment. If we know for sure we did our best *a priori*, and we have a formal guarantee before deployment, then we can more easily trace what went wrong if a crash occurs, because at least one of our assumptions about the system was wrong.

**Key Questions For Authors:**

* I'm a bit confused about how collision detection is actually implemented to generate the training data for the RL part of the method. I guess this comes out of the simulator, but I think the appendix should have more clear information about the vehicle dynamics and collision detection (and simulation?). I don't believe this simulator has any meaningful collision physics, right?

* I really want to hear the authors' reasoning about why this problem makes sense to study in this way.  There are Waymo and Tesla cars with human-in-the-loop corrections already deployed as products, but academics don't generally have access to that scale of data or real-world driving. Do the authors anticipate that those companies are not doing this type of IL+RL solution, and why? What would convince people at Waymo to use the results of this paper (or any recent learning for driving paper)?

**Limitations:**

I don't see any limitations subsection in the paper or explicit callout of limitations. I think that would be very helpful to add.

**Strengths And Weaknesses:**

Strengths:
* The paper addresses an important problem of how to correct neural planners; this is especially important for imitation learning
* The paper is well-written and easy to read, and the figures are clear and nice
* The results look good on a reasonable choice of dataset
* Collision detection for robotics in general is often a key slowdown, so it is a nice idea to train a model to do this

Weaknesses:
* I am very concerned that this method doesn't compare against any standard motion planning baselines... this specific problem, for autonomous driving, has been thoroughly studied with standard motion planning and controls approaches to correct learning in the loop [R1,R2,R3]
* I think the paper would be much stronger if this approach were tested on, for example, manipulation problems against other learning-based correction baselines [R4,R5]

References
[R1] Shao, Y.S., Chen, C., Kousik, S. and Vasudevan, R., 2021. Reachability-based trajectory safeguard (rts): A safe and fast reinforcement learning safety layer for continuous control. IEEE Robotics and Automation Letters, 6(2), pp.3663-3670.

[R2] Pek, C. and Althoff, M., 2020. Fail-safe motion planning for online verification of autonomous vehicles using convex optimization. IEEE Transactions on Robotics, 37(3), pp.798-814.

[R3] Leung, K., Veer, S., Schmerling, E. and Pavone, M., 2023, May. Learning autonomous vehicle safety concepts from demonstrations. In 2023 American Control Conference (ACC) (pp. 3193-3200). IEEE.

[R4] Nakamura, K., Peters, L. and Bajcsy, A., 2025. Generalizing safety beyond collision-avoidance via latent-space reachability analysis. arXiv preprint arXiv:2502.00935.

[R5] Ak, A.C., Aksoy, E.E. and Sariel, S., 2023. Learning failure prevention skills for safe robot manipulation. IEEE Robotics and Automation Letters, 8(12), pp.7994-8001.

---

> ### Author Rebuttal · Authors · 2026-03-31
>
> ## Q1. How collision detection is implemented to generate the training data for RL.
> Unlike the cited control/safety-filtering works, we do not have access to a known dynamics simulator. Instead, as described in Section 3.2, we train a reactive multi-agent world model and use it as the simulator to collect training data for RL training. The policy predicts the ego vehicle’s next position, then the world model generates future trajectories of surrounding agents (up to 32 agents).
>
> Collision and progress are computed from the predicted ego trajectory and the generated-agents' trajectories. So, we not only model the ego trajectory, but also the reactive behavior of surrounding agents, rather than relying on a pre-defined simulator. And we believe the multi-agent system and world model trained on large-scale data is fundamentally different from the manipulation tasks or the tasks in the motion planning safety filtering paper.
> ## Q2. The paper doesn't compare against any standard safe motion planning baselines.
> These are important related works, and we will discuss them more clearly in the revision. However, they address substantially different problems and are therefore not directly comparable as primary baselines.
>
> First, many reachability, safety-filtering methods assume known or structured system dynamics, explicit safety geometry, or reachable/invariant set computation with strong assumptions on the dynamics. In contrast, our setting focuses on large-scale interactive autonomous driving with large real-world datasets and closed-loop evaluation, where complex surrounding-agent dynamics are unavailable and require multi-agent modeling.
>
> Second, these methods are also developed in very different evaluation settings from ours. Our experiments are conducted on WOMD/Waymax and nuPlan, which are large-scale closed-loop driving benchmarks with reactive and non-reactive evaluation, long horizons, and complex multi-agent interactions. In this setting, methods based on explicit reachable-set computation, online convex verification, or low-dimensional control structure are not straightforwardly scalable or directly comparable. For this reason, we compare with modern learning-based closed-loop driving planners trained and evaluated on the same benchmarks, which we believe, combined with ablation studies, provides the most informative comparison for testing whether self-correction improves realistic learned planners.
>
> Overall, to the best of our knowledge, this is the first work to introduce an explicit self-correction mechanism with a correction trace into learning-based autonomous driving and to demonstrate that it has performance improvements in large-scale data. We will revise the paper to provide a clearer discussion of these related works and their relationship to our setting.
> ## Q3. Can the method be tested on a control problem?
> We believe applying our algorithm to manipulation/control tasks is not very meaningful. Similar to recent large-scale learning-based motion planning methods for autonomous driving [1,2,3], which are also not tested in the manipulation tasks, our approach is specifically designed for autonomous driving, with components tailored to multi-agent interaction, maps, and navigation. In addition, our method relies on a world model for reactive multi-agent behavior, whereas the control/manipulation tasks in R4 and R5 are single-agent settings and are therefore fundamentally different. Our focus is on making self-correction work in a multimodal, multi-agent, learning-based planning/navigation setting, where the interaction complexity and behavioral uncertainty go well beyond what is typically captured in classical control problems.
>
> [1] Plan-R1: Safe and Feasible Trajectory Planning as Language Modeling
>
> [2] CarPlanner: Consistent Auto-regressive Trajectory Planning for Large-scale Reinforcement Learning in Autonomous Driving
>
> [3] Diffusion-Based Planning for Autonomous Driving with Flexible Guidance
>
> ## Q4. Given that companies like Waymo have human-in-the-loop driving data, and similar IL+RL approaches are already used in industry, what evidence would make this paper convincing for deployment?
> Human-in-the-loop correction is already used in industry, but they still require frequent intervention. Our method can improve safety by enabling the planner to self-correct before execution, thereby reducing unsafe actions and potentially reducing human intervention.
>
> Second, our contribution is not the IL+RL framework itself; we introduce self-correction into motion planning, enabling the planner to revise unsafe actions before execution. Indeed, human-in-the-loop data from Waymo could help the training, but collecting such data is costly, and our method helps reduce that burden. We believe this idea could be easily incorporated and generalized into real autonomous driving systems, with more data and a more powerful foundation model, to improve safety through more advanced safety reasoning.

---

> > ### Author Rebuttal · Reviewer_MLyu · 2026-03-31
> >
> > I thank the authors for carefully considering my comments and providing thoughtful rebuttal responses.
> >
> > Q1/2. I disagree that training this type of world model is substantially different enough from existing simulation frameworks to invalidate comparisons against existing safety-focused baselines. At the end of the day, the proposed system generates trajectories of the ego and other agents... so the world model is a simulator.
> >
> > Q3. I partially agree with this rebuttal; the comparison on manipulation is perhaps not relevant for this work, but I encourage the authors to consider it for future work. It seems like the proposed method could be used for multi-agent manipulation. From a research perspective, compared to autonomous driving, it feels like a world model is much more necessary for collaborative manipulation of a deformable object, where the other agents *and* the object dynamics are hard to model but can be learned from data.
> >
> > Q4. I am not very convinced by this answer. Self-correction in motion planning is not a new idea, nor is incorporating it into a multi-agent world model / simulator, nor into an IL policy (a few variations are here: [R6,R7,R8]). Furthermore, I do not feel that saying this method is safer (but still has tons of collisions) is very convincing, because of the existence (for nearly a decade!) of formal methods with strict safety guarantees, which have already been folded into learning frameworks (see earlier review citations). Why should a company, which has to afford liability insurance, use something that is distinctly less safe?
> >
> > Perhaps the more important questions to answer going forward are, (1) what are the gaps with incorporating formal safety methods (as opposed to approximate safety losses) into autonomous driving world models / planners? and (2) how do we get the same safety rate as existing learning-based planners with much less data or smaller modular architectures? I recognize that this paper as-is does not try to answer these questions, but I really encourage the authors to think about these directions for the future.
> >
> > P.S. I found a contemporaneous paper that the authors might enjoy reading [R9]
> >
> > **References**
> >
> > [R6] Ke, L., Zhang, Y., Deshpande, A., Srinivasa, S. and Gupta, A., 2023. Ccil: Continuity-based data augmentation for corrective imitation learning. arXiv preprint arXiv:2310.12972.
> >
> > [R7] Zhu, M., She, H., Si, W. and Li, C., 2024, August. Lightweight imitation learning algorithm with error recovery for human direction correction. In 2024 29th International Conference on Automation and Computing (ICAC) (pp. 1-6). IEEE.
> >
> > [R8] Antotsiou, D., Ciliberto, C. and Kim, T.K., 2021, May. Adversarial imitation learning with trajectorial augmentation and correction. In 2021 IEEE International Conference on Robotics and Automation (ICRA) (pp. 4724-4730). IEEE.
> >
> > [R9] Ma, E., Zhou, L., Tang, T., Zhang, J., Jiang, J., Zhang, Z., Han, D., Zhan, K., Zhang, X., Lang, X. and Sun, H., 2026, March. CorrectAD: A Self-Correcting Agentic System to Improve End-to-end Planning in Autonomous Driving. In Proceedings of the AAAI Conference on Artificial Intelligence (Vol. 40, No. 10, pp. 7755-7763).

---

> > > ### Author Response · Authors · 2026-04-02
> > >
> > > ## Q1/2 Formal safety methods
> > > We agree these are principled and related methods. But their evaluation/problem settings do not match ours, so they are not directly comparable.
> > > 1. **They are developed for more abstract or simplified environments and are primarily designed around safety guarantees, while we study a more realistic, complex, and data-driven problem that evaluates general planning ability beyond safety.**
> > > R1 and R3 consider static obstacles or two-vehicle interaction, while WOMD and nuPlan have rich and more diverse multi-agent traffic scenes. Also, prior methods abstract away many aspects of realistic driving and are primarily designed to ensure safety. In contrast, ours and real-world driving are multi-objective problems that require general planning ability, with designed components to handle multi-agent interaction, map/navigation compliance, and comfort improvement, among others. **These are evaluated using the planning scores in the nuPlan benchmark, and these aspects of planning are not considered in previous work**. In addition, many of them are evaluated only on simple tasks like highway lane changing, but our benchmarks include more realistic and challenging scenarios like intersections, roundabouts, etc.
> > > 2. **Scaling them to our setting requires substantial redesign.** Methods like R2 still require additional designs that are not part of the original method, including navigation/complex-map integration, multi-agent aggregation, and occupancy-set design, which are important design choices. In addition, the safety layer in these methods cannot be directly plugged into current end-to-end planning baselines without redesign. But incorporating them (also to multi-agent manipulation tasks) is a promising research direction (see Q5.2).
> > > 3. Incomparable isn't due to the difference in world model/simulator, and the world model is not our contribution.
> > > ---
> > > ## Q4 Compared with correction methods [R6-9]
> > >
> > > **We are fundamentally different and not a variant of [R6-9]**. They correct the training data to provide further training signals, and R7 has human-corrected training data, which we do not have. Instead, we perform self-correction during inference.
> > >
> > > Our self-correction mechanism with propose–evaluate–correct in an autoregressive way is novel and fundamentally different from prior so-called self-correction and formal-safety methods. **We are the first to propose such inference self-correction encoded an autoregressive NTP framework for driving using an explicit correction/thinking trace**, and the novelty has been acknowledged by other reviewers.
> > >
> > > ---
> > > ## Q5 What are the gaps in incorporating formal safety methods
> > > 1. **Strong assumptions and simpler environments [R1, R3]. Redesign is needed.** Many of them consider more abstract settings, assuming known dynamics, or hand-designed parametric structure for static or 1/2 agents' interaction. So, extending them to complex traffic requires redesign of the system/parametric models. Also, in realistic driving, safety is critical, but so are progress, drivable-area/navigation compliance, and comfort. These aspects are not considered in many formal-safety methods, so a redesign is needed to incorporate complex information (maps, navigation, etc.) and integrate them properly with the safety layer.
> > > 2. **The research question is how to improve safety without degrading performance on other important planning objectives, and how to integrate/design safety layers with guarantees that are naturally aligned with broader objectives rather than simply plugging them into the planner to only ensure safety, which can be coarse and conservative without specific design.** Formal safety methods have not yet been studied in large-scale, data-driven and realistic driving benchmarks like ours, making this an important future direction. Our self-correction mechanism is a step in this direction, and incorporating formal guarantees is a promising future direction.
> > > ---
> > > ## Q6 Safer with less data or smaller architectures
> > > **Our safety claim is not overstating, and industrial-level and formal safety methods cannot eliminate collisions**. Our safety claim does not assert a collision-free planner, nor do we compare against industrial-level models or formal-safety methods, which do not fall into our baseline as mentioned, and whose performance on different planning goals remains unclear. **Compared with SOTA data-driven learning-based methods [1,2,3], we show that our self-correction mechanism reduces collisions while maintaining similar performance on other metrics and improving the overall planning score, suggesting safer behavior.**
> > >
> > > Further, the industry is now switching to the data-driven end-to-end planning method. So we believe this paper advances the domain by showing that, under fixed data and model budgets, self-correction improves the safety–performance trade-off.
> > >
> > > **Due to space limit, we cannot elaborate on all points, but we can clarify any part that remains unclear.**

---

### Decision · Program_Chairs · 2026-04-30

**Decision:**

Accept (regular)

**Comment:**

I have read most of the reviews and the authors' responses. The positive comments mainly focus on the fact that it is indeed a highly complete and effective neural planner solution for autonomous driving, supported by extensive experimental results, with a very clear structure and writing. The negative comments primarily point out the lack of formal safety guarantees, and that the entire framework contains too many components, which raises concerns about its usability and makes it difficult to isolate the effectiveness of the core method. Overall, I consider this a solid paper that points out a practical path, and it can be accepted if there is room in the program; however, it is indeed not elegant enough from an academic perspective, and its scope of impact is relatively narrow and specialized.